

# The Different Dynamic Influences of Typhoon Kalmaegi on two Pre-existing Anticyclonic Ocean Eddy

Yihao He[1], Xiayan Lin [1,2,*], Guoqing Han [1], Yu Liu [1,3] and Han Zhang [2,3,*]

1 Marine Science and Technology College, Zhejiang Ocean University, Zhoushan 316022, China;

2 State Key Laboratory of Satellite Ocean Environment Dynamics, Second Institute of Oceanography,

Ministry of Natural Resources, Hangzhou 310012, China;

3 Southern Marine Science and Engineering Guangdong Laboratory (Zhuhai), Zhuhai 519082, China

*Correspondence: Xiayan Lin (linxiayan@zjou.edu.cn) and Han Zhang (zhanghan@sio.org.cn)

**Abstract:** Using multi-source observational data and GLORYS12V1 reanalysis data, we conducted a comparative analysis of different responses of two warm eddies, AE1 and AE2 in the northern South China Sea to Typhoon Kalmaegi during September 2014. The findings of our research are as follows: (1) For horizontal distribution, the area and the sea surface temperature (SST) of AE1 and AE2 decreased by about 31% (36%) and 0.4 ℃ (0.6 ℃).The amplitude, Rossby number ($R_o$) and eddy kinetic energy (EKE) of AE1 increased by 1.3 cm, $1.4×10^{-2}$ and 107.2 $cm^2$ $s^{-2}$ after the typhoon, respectively, while AE2 weakened and the amplitude, vorticity and EKE decreased by 3.1 cm, $1.6×10^{-2}$ and 38.5 $cm^2$ $s^{-2}$, respectively. (2) In vertical direction, AE1 demonstrated enhanced convergence, leading to an increase in temperature and a decrease in salinity above 150 m. The response below the mixing layer depth (MLD) was particularly prominent (1.3 ℃). In contrast, AE2 experienced cooling and a decrease in salinity above the MLD. Below the MLD, it exhibited a subsurface temperature drop and salinity increase due to the upwelling of cold water induced by the suction effect of the typhoon. (3) The disparity in the responses of the two warm eddies can be attributed to their different positions relative to Typhoon Kalmaegi. Warm eddy AE1, with its center located on the left side of the typhoon's path, experienced a positive work effect as the typhoon passed by. This induced a strong negative wind stress curl and triggered a negative Ekman pumping velocity (EPV), further enhanced by the converging sinking of the upper warm water, thereby strengthening AE1. On the other hand, warm eddy AE2, situated closer to the center of the typhoon, weakened due to the cold suction caused by the strong positive wind stress curl in the typhoon's center. These findings underscore the importance of relative positions of eddies in their interactions with typhoons.



**1. Introduction**

Typhoons, as they traverse the vast ocean, interact with oceanic mesoscale processes, particularly

with mesoscale eddies, representing a crucial aspect of air-sea interaction (Shay and Jaimes, 2010; Lu et
al., 2016; Song et al., 2018; Ning et al., 2019; Sun et al., 2023). The South China Sea (SCS) experiences
an average of six typhoons passing through each year (Wang et al., 2007). Meanwhile, the northern part
of the South China Sea (NSCS) encounters frequent eddy activities due to the influence of the Asian
monsoon, intrusion of the Kuroshio Current, and the impact of topography (Xiu et al., 2010; Chen et al.,
2011). This unique setting offers an exceptional opportunity to investigate the generation, evolution, and
termination of mesoscale eddies and their interaction with typhoons.

On one hand, tropical cyclones (TCs) derive their development and sustenance energy from the ocean.

Pre-existing mesoscale eddies play a crucial role in the feedback mechanism between the ocean and TCs.
Cyclonic eddies (cold eddies) enhance the sea surface cooling effect under TC conditions, resulting in
TCs weakening, due to their thermodynamic structure and cold-water entrainment processes that reduce
the heat transfer from the sea surface to the typhoon through air-sea interaction(Ma et al., 2017; Yu et
al., 2021). In contrast, anticyclonic eddies (warm eddies) suppress this cooling effect, leading to TC
intensification (Shay et al., 2000; Walker et al., 2005; Lin et al., 2011; Wang et al., 2018). Warm eddies
have a thicker upper mixed layer, which stores more heat. When a typhoon passes through a warm eddy,
it increases sensible heat and water vapor in the typhoon's center, which are closely related to the
typhoon's intensification (Wada and Usui, 2010; Huang et al., 2022). Furthermore, the downwelling
within warm eddies hinders the upwelling of cold water, reducing the apparent sea surface cooling caused
by the typhoon. This weakens the oceanic negative feedback effect and helps to sustain or even strengthen
the typhoon's development.

On the other hand, TCs can induce various oceanic processes such as local advection, vertical mixing,

and upwelling, leading to a decrease in sea surface temperature (SST). The cooling effect typically ranges
from 2-4°C and can reach up to 10°C under extreme conditions (Price, 1981; Wu et al., 2011; Han et al.,
2012). The distribution of typhoon wind stress and variations in vertical mixing cause different cooling
patterns on both sides of the typhoon track in the upper ocean. Generally, the right side exhibits stronger
cooling of the SST in the northern hemisphere (Stramma et al., 1986; Vincent et al., 2012; Mei et al.,
2015; Mitarai and Mcwilliams, 2016).TCs also have a notable impact on the intensity, size, and



movement of mesoscale eddies. In general, TCs strengthen cold eddies and can even lead to the formation
of new cyclonic eddies in certain situations (Sun et al., 2014), while TCs accelerate the dissipation of
anticyclonic eddies (Zhang et al., 2020). The interaction between TCs and eddies directly affects the
local upper ocean structure and circulation system.
The strengthening effect of TCs on cold eddies is related to the positions between cold eddies and
TCs, the intensity of eddies, and TC-induced geostrophic response (Lu et al., 2016; Yu et al., 2019; Lu
et al., 2023). Cyclonic eddies on the left side of the typhoon track were more intensely affected by the
typhoon than eddies on the right side, and eddies with shorter lifespans or smaller radii are more
susceptible to the influence of typhoons. The dynamic adjustment process of eddy and the upwelling
induced by the typhoon itself leads to changes in the three-dimensional structure of the cyclonic eddies,
including ellipse deformation and re-axisymmetrization on the horizontal plane, resulting in eddy
intensification. The presence of cold eddies not only exacerbates the sea surface cooling in the post-
typhoon cold eddy region but also accompanies a decrease in sea level anomaly (SLA), deepening of the
mixed layer, a strong cooling in the subsurface, increased chlorophyll-a concentration within the eddy,
and substantial increases in eddy kinetic energy (EKE) and available potential energy (Shang et al., 2015;
Liu and Tang, 2018; Li et al., 2021; Ma et al., 2021).
Generally, typhoons lead to a reduction of warm eddies, while the sea surface cooling is not
significant, typically within 1℃. However, there is a noticeable cooling and increased salinity in the
subsurface layer, accompanied by an upward shift of the 20℃ isotherm, a decrease in heat and kinetic
energy (Lin et al., 2005; Liu et al., 2017; Huang and Wang, 2022). Lu et al. (2020) proposed that typhoons
primarily generate potential vorticity input through the geostrophic response. When a typhoon passes
over an eddy, there is a significant positive wind stress curl within the typhoon's maximum wind radius,
which induces upwelling in the mixed layer due to the divergence of the wind-driven flow field. This
upward flow compresses the thickness of the isopycnal layers below the mixed layer, resulting in a
positive potential vorticity anomaly. By analyzing the time series of ocean kinetic energy, available
potential energy (APE), vorticity budget, and potential vorticity (PV) budget, Rudzin and Chen (2022)
found that the positive vertical vorticity advection caused the TC to eliminate the warm eddy from bottom
to top after passing through. Under the interaction of the strong TC wind stress in the eye area of the
typhoon and the subsurface ocean current field, the early-onset of a near-inertia wake caused the
disappearance of the warm eddy. However, the projection of TC wind stress onto the eddy and the relative



position of the warm eddy to the typhoon can lead to different responses. According to the classical
description of TC-induced upwelling, strong upwelling occurs within twice the maximum wind radius
of the typhoon center, while weak subsidence exists in the vast area outside the upwelling region (Price,
1981; Jullien et al., 2012). The warm eddy located directly beneath the typhoon's path weakens due to
the cold suction caused by the typhoon's center. However, for warm eddies located beyond twice the
maximum wind radius, they are influenced by the typhoon's wind stress curl and the downwelling within
the eddy itself, resulting in the convergence of warm water in the upper layers of the eddy, an increase
in mixed layer thickness, and an increase in heat content, leading to a warming response to the typhoon
(Jaimes and Shay, 2015).

Previous studies on the interaction between warm eddies and typhoons have primarily focused on the

enhancing impact of warm eddies on typhoons. However, there has been relatively limited exploration
of different responses exhibited by warm eddies under the influence of typhoons. In this study, in-situ
measurements, remote sensing data, and GLORYS12V1 reanalysis data are utilized to investigate distinct
responses of two warm eddies to typhoon Kalmaegi in the NSCS. Section 2 provides an overview of the
data and methods utilized in this research. Section 3 analyzes the physical parameters of warm eddies,
vertical temperature and salinity variations, and explores the different responses of warm eddies both
inside and outside the typhoon affected region. Section 4 offers a comprehensive discussion and Section
5 gives a summary.
**2. Data and Methods**
**2.1. Data**

The six-hourly best-track typhoon datasets were obtained from the Joint Typhoon Warning Center
(JTWC, http://www.usno.navy.mil/JTWC, last access: 3 February, 2021), the Japan Meteorological
Agency    (JMA,https://www.jma.go.jp/jma/jma-eng/jma-center/rsmc-hp-pub-eg/besttrack.html,    last
access:    3    February,    2021),    and    the    China    Meteorological    Administration    (CMA,
http://tcdata.typhoon.gov.cn, last access: 3 February, 2022). The data contained the tropical cyclone
center locations, the minimum central pressure, maximum sustained wind speed, and intensity category.
The translation speed of typhoons was calculated by dividing the distance travelled by each typhoon
within a 6-hour interval by the corresponding time. In this paper, typhoon Kalmaegi and tropical storm
Fung-wong were studied (Fig. 2).

The daily Sea Level Anomaly (SLA) and geostrophic current data provided by Archiving, Validation,
and    Interpretation    of    Satellite    Data    in    Oceanography    (AVISO)    product    (CMEMS,



https://marine.copernicus.eu/, last access: 14 Febururay, 2022). This dataset combines satellite data from
Jason-3, Sentinel-3A, HY-2A, Saral/AltiKa, Cryosat-2, Jason-2, Jason-1, T/P, ENVISAT, GFO, and
ERS1/2. The spatial resolution of the product is $1/4° × 1/4°$, the period from 1 September to 30 September
2014 was used.
The daily Sea Surface Temperature (SST) data used in this study is derived from the Advanced Very
High Resolution Radiometer (AVHRR) product data provided by the National Oceanic and Atmospheric
Administration (NOAA). The data is obtained from the Physical Oceanography Distributed Active
Archive Center (PODAAC) at the NASA Jet Propulsion Laboratory (JPL)
(ftp://podaac.jpl.nasa.gov/documents/dataset_docs/avhrr_pathfinder_sst.html, last access: 16 March,
2022). The spatial resolution of the data is $1/4° × 1/4°$.
Argo data, including profiles of temperature and salinity from surface to 2000 m depth are obtained
from the real-time quality-controlled Argo data base (Euro-Argo, https://dataselection.euro-argo.eu/, last
access: 4 April, 2022). We selected Argo float number 2901469, situated in an ocean anticyclonic eddy
and in close proximity to typhoon Kalmaegi, both before and after the typhoon's passage in 2014. Profiles
of this Argo were also used to validate the vertical distribution of temperature and salinity from
GLORYS12V1.
For this study, we also utilized in situ data from a cross-shaped array consisting of five stations,
comprising five moored buoys and four subsurface moorings (refer to Fig. 2). More specific information
can be found in Zhang et al. (2016). To investigate the impact of the typhoon on a warm eddy, we selected
the temperature and salinity data from Station 5, situated along the left track of Kalmaegi.
The wind speed data was sourced from the European Centre for Medium-Range Weather Forecasts
(ECMWF) ERA-Interim reanalysis assimilation dataset (https://apps.ecmwf.int/datasets/data/interim-
full-daily/levtype=sfc/ , last access: 5 January, 2023). This dataset was widely used for weather analysis
and numerical forecasting. The wind field data used in this study primarily focused on the reanalysis data
of surface winds at a height of 10 meters above sea level for tropical cyclones. The selected data had a
spatial resolution of $1/4° × 1/4°$ and a temporal resolution of 6 hours, with four updates per day (00:00,
06:00, 12:00, and 18:00 UTC). The data utilized corresponds to September 2014.
The Global Ocean Reanalysis Product GLOBAL_REA- NALYSIS_PHY_001_030 (GLORYS12),
provided by the Copernicus Marine Environment Monitoring Service (CMEMS,
https://marine.copernicus.eu/, last access: 23 March, 2022) was used in this study too. This reanalysis
product utilized the NEMO 3.1 numerical model coupled with the LIM2 sea ice model, and forced with
ERA-Interim atmospheric data. The model assimilated along-track altimeter data from satellite
observations (Pujol et al., 2016), satellite sea surface temperature data from AVHRR, sea ice
concentration from CERSAT (Ezraty et al., 2007), and vertical profiles of temperature and salinity from
the CORAv4.1 database (Cabanes et al., 2012). The temperature and salinity biases were corrected using
a 3D-VAR scheme. The horizontal resolution is $1/12° × 1/12°$, and it has 50 vertical levels. The
temperature and salinity during 1 September to 30 September 2014 was chosen to study.
GLORYS12V1 is a widely used and applicable dataset, to evaluate its temperature profiles, the Argo
profiles and in-situ data of Station 5 were compared (Fig. 1). The GLORYS12V1 data exhibit good





agreement with Argo profiling floats, the maximum difference between them is less than 0.2℃. However,
there are some discrepancies between the GLORYS12V1 and the Station 5 data, with the largest
difference occurring at the depths of 30 m (mixed layer) and 78 m (thermocline), both differing by 0.6℃,
while below 150 m, the difference is quite small. This may be because the vertical resolution of upper
100 m in Argo profile is 5 m, but the vertical interval of Station 5 is 20 m, it is sparser. Therefore, the
large deviations exist at mixed layer and thermocline during the typhoon in in-situ data of Station 5.
Overall, GLORYS12V1 reproduces the observed ocean temperature accurately, it is reasonable to use it
to investigate the vertical feedback of the ocean by typhoon Kalmaegi.

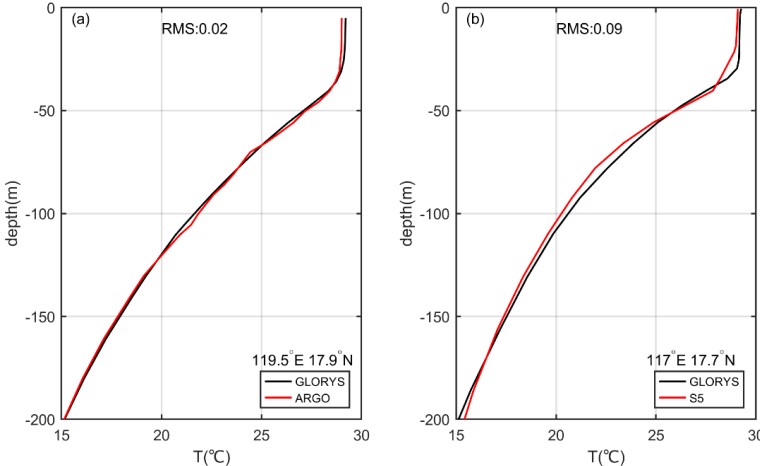


**Figure 1.** Evaluation of GLORYS12V1 data performance during September 2014. **(a)** Vertical monthly mean
temperature within the anticyclonic eddy AE2 (119.5°E 17.9°N) as measured by Argo float 2901469. **(b)**
Comparison of vertical monthly mean temperature recorded at Station 5 (117°E 17.7°N).
**2.2. Methods**
Vorticity is a vector that characterizes the local rotation within a fluid flow. Mathematically, it is
defined as the curl of the velocity vector. In most cases, when referring to vorticity, it specifically pertains
to the vertical component of the vorticity. It is calculated from:
$$\zeta = \frac{\partial v}{\partial x} - \frac{\partial u}{\partial y} \; . \tag{1}$$

$u$ and $v$ are the zonal (eastward) and meridional (northward) geostrophic velocities, respectively. They
are derived from altimeter sea level anomaly data ($\eta$):
$$u = -\frac{g}{f}\frac{\partial \eta}{\partial y} \; , v = \frac{g}{f}\frac{\partial \eta}{\partial x} \; . \tag{2}$$

Here, $g$ is the acceleration of gravity, $f$ is the Coriolis frequency. Vorticity is considered a
fundamental characteristic of mesoscale eddies, positive vorticity signifies cyclonic eddies, while
negative vorticity indicates anticyclonic eddies.



The Rossby number (Ro) is a dimensionless number describing fluid motion, and it is the ratio of
relative vorticity to planetary vorticity, reflecting the relative importance of local non-geostrophic motion
to large-scale geostrophic motion. The larger the Rossby number, the stronger the local non-geostrophic
effect, and the definition of this parameter is:
$$R_o = \frac{\zeta}{f} \ . \tag{3}$$

Eddy Kinetic Energy (EKE) is a measure of the energy associated with mesoscale eddies, which
indicates the intensity of eddies. It is typically calculated using the anomalies of the geostrophic velocity:
$$EKE = \frac{1}{2}(u'^2 + v'^2) \ , \tag{4}$$

where u' represents the anomaly of the geostrophic zonal (eastward) velocity, v' represents the anomaly
of the meridional (northward) velocity.
To evaluate the impact of a typhoon on an anticyclonic eddy, the calculation begins with determining
the wind stress:
$$\vec{\tau} = \rho_a C_d U_{10} \overrightarrow{U_{10}} \ , \tag{5}$$

where $\rho_a$ is the air density, assumed to be a constant value of 1.293 kg m$^{-3}$, $U_{10}$ represents the 10-
meter wind speed. And $C_d$ is the drag coefficient at the sea surface (Oey et al., 2006):
$$C_d \times 1000 = \begin{cases} 1.2 & U_{10} \leq 10m \ s^{-1} \\ 0.49 + 0.65U_{10} & 11 \leq U_{10} < 19m \ s^{-1} \\ 1.364 + 0.234U_{10} - 0.00023158U_{10}{}^2 & 19 \leq U_{10} \leq 100m \ s^{-1} \end{cases} \ . \tag{6}$$

The wind stress curl is calculated by (Kessler, 2006):
$$curl(\vec{\tau}) = \frac{\partial \tau_y}{\partial x} - \frac{\partial \tau_x}{\partial y} \ , \tag{7}$$

where $\tau_x$ and $\tau_y$ are the eastward and northward wind stress vector components, respectively. The curl
represents the rotation experienced by a vertical air column in response to spatial variations in the wind
field.
The Ekman pumping velocity (EPV) represents the ocean upwelling rate, which can be used to study
the contribution of typhoons to regional ocean upwelling. Positive means upwelling, negative represents
downwelling:
$$EPV = curl(\frac{\vec{\tau}}{\rho f}) \ , \tag{8}$$

where the wind stress is obtained from Eq. (7), $\rho$ is seawater density, the value is 1025 kg m$^{-3}$, and $f$
is the Coriolis frequency.
The buoyancy frequency is a measure of the degree to which water is mixed and stratified. In a stable
temperature stratification, the fluid particles move in the vertical direction after being disturbed, and the
combined action of gravity and buoyancy always makes them return to the equilibrium position and
oscillate due to inertia. The frequency of oscillation is the floating frequency (N). When $N^2 < 0$, the



water is in an unstable state. The larger N is, the lower the degree of mixing and the higher the degree of
stratification:
$$N = \sqrt{-\frac{g}{\rho}\frac{\partial \rho}{\partial z}}.$$
(9)

Where $\rho$ is seawater density, $g$ is the acceleration of gravity, and z is the depth.
**3. Results**
**3.1. Typhoon and pre-existing eddies in the NSCS**
**3.1.1. Track of typhoon Kalmaegi and tropical storm Fung-wong**
Tropical cyclone Kalmaegi strengthened into a typhoon by 1200 UTC on 13 September and emerged
over the warm waters of the Northern South China Sea (NSCS) by 1500 UTC on 14 September, with
maximum sustained winds of 33 m s$^{-1}$ (Fig. 2-3). During this period, the NSCS experienced
predominantly weak wind shear (Fig. 4a) and was characterized by multiple anticyclonic warm eddies
(Fig. 2). Subsequently, typhoon Kalmaegi underwent two rapid intensification phases between 15 and
16 September (Fig. 4c-f). The first intensification occurred at 0000 UTC on 15 September, propelling
Kalmaegi to category 1 status with surface winds surpassing 35 m s$^{-1}$. By 1200 UTC on 15 September,
Kalmaegi experienced a second, even more rapid intensification, with winds reaching 40 m s$^{-1}$ in less
than 12 hours. Throughout this intensification stage, Kalmaegi encountered two warm eddies:
anticyclonic eddy AE1, located to the left of the typhoon's path (Fig. 3), which had a lifespan of 105 days
from 26 June to 8 October and was positioned at 17°N-20°N, 113°E-116°E, and AE2, precisely
intersecting with the typhoon's trajectory, which had a lifespan of 89 days from 24 August to 20
November and was located at 17°N -19°N, 118°E -120°E. Kalmaegi made landfall on Hainan Island at
0300 UTC on 16 September, with a minimum central pressure of 960 hPa and maximum wind speed of
40 m s$^{-1}$. After landfall, Typhoon Kalmaegi gradually weakened and dissipated. During it across the
NSCS, the five mooring stations were affected. Stations 1 and 4 were on the right side of Typhoon
Kalmaegi's track, while Stations 2 and 5 were on the left side. Unfortunately, the wire rope of the buoy
at Station 3 was destroyed by Kalmaegi, resulting in missing data from 15 September. Among the stations,
Station 5 is on the left of typhoon track and outside AE2, so its data is used in our study.
Tropical storm Fung-wong initially moved quickly in a northwest direction after formation. On 19
September, it entered the Luzon Strait and slowed down. It made landfall in Taiwan on the 21 September
and subsequently landed in Zhejiang on the 22 September before gradually dissipating. When crossing
the Luzon Strait at 1200 UTC on 19 September, anticyclonic eddy AE2 was on the left side of Fung-
wong with a distance of just over 100 km from its centre.

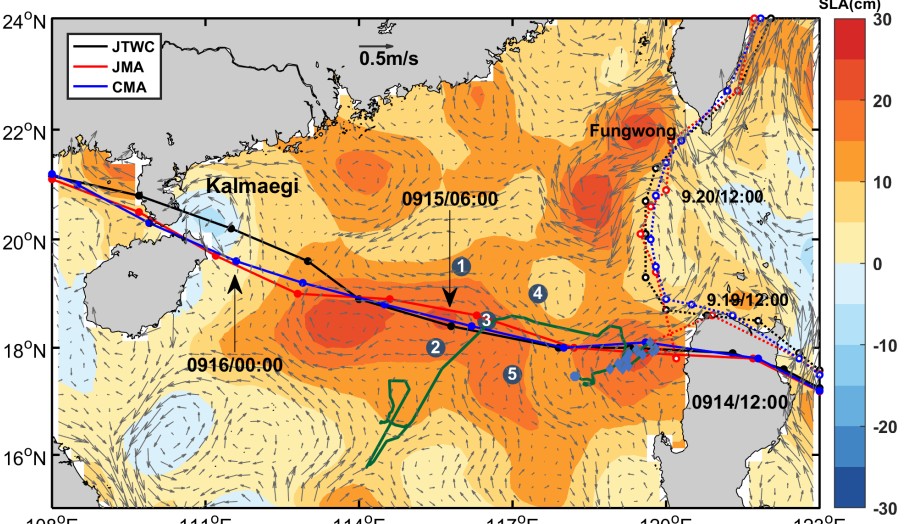

**Figure 2.** The tracks of typhoon Kalmaegi (solid lines with dots) and tropical storm Fung-wong (dashed lines with hollow dots) as provide by the Joint Typhoon Warning Center (JTWC, black), Japan Meteorological Agency (JMA, red), and China Meteorological Administration (CMA, blue). The colour shading represents the sea surface level anomaly on 13 September, 2014, while the gray arrows illustrate the geostrophic flow field. The numbered blue dots represent the positions of the five buoy/mooring stations, the green line illustrates the trajectory of Argo 2901469, and the blue diamonds mark the positions of Argo 2901469 inside the eddy AE2 from 26 August 2014 to 25 October 25, 2014.

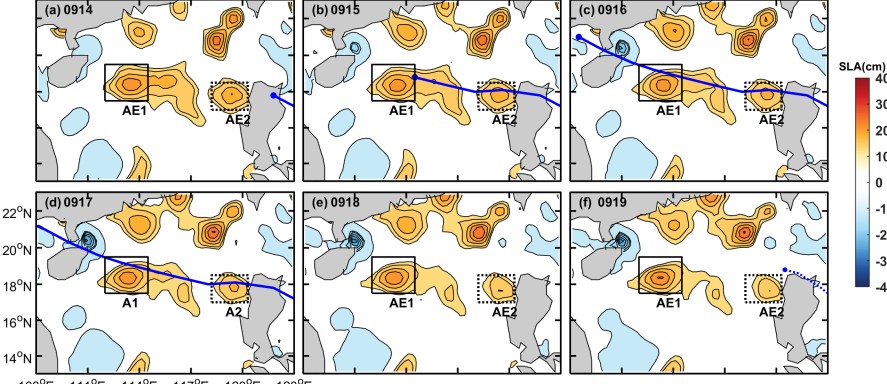

**Figure 3.** The variations in sea level anomaly before and after typhoon Kalmaegi moved over the anticyclonic eddies AE1 and AE2 between 14 September and 19 September **(a-f)**. The black solid rectangle represents the area of AE1, while the black dashed rectangle represents the area of AE2. The blue solid line depicts the path of typhoon Kalmaegi, while the blue dotted line in **(f)** is the path of tropical storm Fung-wong (best-track data sourced from CMA).



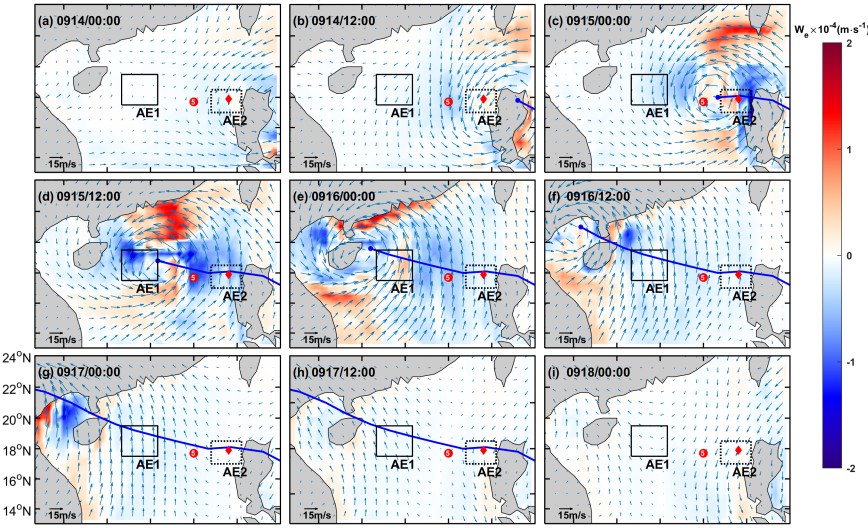

**Figure 4.** Ekman Pumping Velocity (EPV) from 14 September to 18 September **(a-i)**. The color represents the EPV, the blue solid line is the path of Kalmaegi, the red dot and diamond are the positions of Station 5 and Argo 2901469 on 15 September, respectively.

### 3.1.2. Eddy characteristics distribution

Satellite SLA measurements have proven to be highly effective and widely used for identifying and quantifying the intensity of ocean eddies (Li et al., 2014). In Fig. 3, two warm eddies with clear positive (>13 cm) SLA are observed along the typhoon Kalmaegi's track. During the period of 15 to 16 September, the typhoon passed over two warm anticyclonic eddies, AE1 and AE2.Before typhoon, AE1 is the most prominent eddy in the SCS, with an amplitude of 23.0 cm, and a radius of 115.5 km. AE2, located west of Luzon Island, exhibits an amplitude of 21.2 cm, with a radius of approximately 65.5 km. Tracing back to 2 months (figure is not shown), AE1 propagated slowly westward with about 0.1 m s$^{-1}$, while AE2 was generated on 24 August. During 14 to 19 September, the amplitude of AE1 increased 1.3 cm. The area of the AE1 decreased by approximately 31% from $1.3\times10^5$ km$^2$ to $9.1\times10^4$ km$^2$ and split into two eddies. When typhoon Kalmaegi crossed the core of AE2 at 1500 UTC on 14 September, and tropical storm Fung-wong moved over the northeast of AE2 at 1200 UTC on 19 September, the amplitude decreased by 3.1 cm. The area of the AE2 decreased by approximately 36% from $4.2\times10^4$ km$^2$ to $2.7\times10^4$ km$^2$. After 19 September, the influence of the typhoon on the warm eddies gradually diminished.

Because of intense solar radiation in September, the SST in the South China Sea was generally above 28.5℃ prior to the arrival of typhoon Kalmaegi (Fig. 5a). As a fast-moving typhoon, the mean moving speed of typhoon Kalmaegi over 8 m s$^{-1}$, the cooling area and intensity on the right side of the path are larger compared to the left side (Price, 1981). During the passgae of Kalmaegi, the lowest SST on the right side of typhoon decreased to 27.2℃. Even after the typhoon has passed, a cold wake can still be observed on the right side of the path, persisting for over a week (Fig. 5c).





Mesoscale eddies, due to their special thermodynamic structure and varying positions in relation to

the typhoon, can modulate distinct sea surface temperature changes and exhibit different characteristics.
The pre-existing warm eddy AE1 began to cool down before the typhoon reached the NSCS, dropping
to 28.4°C on September 14. Meanwhile, the Ekman Pumping Velocity (EPV) was very small, smaller
than $0.5 \times 10^{-5}$ m s$^{-1}$ in both AE1 and AE2. During 15-16 September (Fig. 4c-f), when the typhoon
traversed the NSCS, the EPV experienced significant changes, the EPV increased to over $1.5 \times 10^{-4}$ m s$^{-1}$
within AE1 and AE2. The positive EPV contributed to the influx of colder subsurface water into the
upper layers, resulting in surface water cooling, while the negative EPV facilitated downwelling and
strengthened the influence of the warm eddies (Jaimes and Shay, 2015), during this period, the mean
SST within AE1 increased slightly to 28.6 °C (Fig. 6a). However, as cooler water from the right side of
the typhoon track was subsequently advected into the AE1 region (Fig. 5c), the SST decreased and
reached 28.0 °C on September 19, which was 0.4°C lower than that before the typhoon. The average sea
temperature drop in AE2 was relatively evident, with SST starting to decline before September 14 and
reaching its lowest point of 28.1°C on September 15, which was 0.6 °C lower than that before the typhoon
(Fig. 6e). On 16 September, the SST within AE2 began to recover, but it started to cool again on 18
September due to the influence of Fung-wong.

Then we compared the Rossby number (Ro) and EKE of AE1 and AE2 before, during and after

typhoon. Before being influenced by the typhoon, the warm eddy AE1 exhibited a more scattered
distribution of negative Ro due to its edge structure, and the EKE values at the eddy boundary were
relatively high (Fig. 5d, g). As the typhoon passed through the eddy, the Ro and EKE of AE1 started to
increase. On 19 September, the average Ro within AE1 reached a value of $-8.2 \times 10^{-2}$, at the same time,
the average EKE increased to its maximum value of 325.0 cm$^2$ s$^{-2}$. It can be observed that the variation
trend of Ro and EKE within the eddy is consistent, increasing from the passage of the typhoon and
starting to recover on 20 September (Fig. 6b-c). This indicates that although the area of the warm eddy
AE1 decreased under the influence of the typhoon, its intensity increased. On the other hand, for warm
eddy AE2, the Ro and EKE both decreased after the typhoon passage, with the Ro decreasing to $-4.5 \times 10^{-2}$
on 17 September and the EKE decreasing to 152.0 cm$^2$ s$^{-2}$ on the 19 September, followed by a recovery
(Fig. 6f-g). Unlike AE1, AE2 weakened in intensity under the influence of the typhoon.

During the passage of the typhoon, the enhanced mixing driven by wind stress and increased vertical

shear result in a deepening of the mixed layer depth (MLD), which further strengthens the mixing
between the deep cold water and the upper warm water (Shay and Jaimes, 2009). To avoid a large part
of the strong diurnal cycle in the top few meters of the ocean, 10 m was set as the reference depth (De
Boyer Montégut, 2004). A 0.5 °C threshold difference from 10 m depth was calculated and defined as
the MLD (Thompson and Tkalich, 2014). Prior to the typhoon passage, the MLD in the AE1 and AE2
regions is deeper (Fig. 5j), the average MLDs of AE1 and AE2 are 32 m and 33 m, respectively. Starting
from September 14th, the MLDs were influenced by typhoon Kalmaegi, with the MLD of AE1 deepening
to 37 m and that of AE2 increasing to 41 m, representing a deepening of 5 m and 8 m, respectively (Fig.
6d, f). At the same time, the MLD on the right side of the typhoon track is also increasing, and the SST
in the corresponding area also drops significantly (Fig. 5l).
Overall, typhoon Kalmaegi likely exerted distinct impacts on the two warm eddies. Despite both AE1
and AE2 experiencing a decrease in their respective areas by approximately one-third, and are
accompanied by deepening of the MLD, the amplitude of sea level anomaly (SLA) within AE1 increased
by 1.3 cm, whereas AE2 witnessed a decrease of about 3.1 cm in its amplitude. Furthermore, the sea
surface temperature (SST), Rossby number and eddy kinetic energy (EKE) within AE1 and AE2
exhibited contrasting patterns. In the following sections, we will delve into the underlying reasons behind
these divergent responses of the two eddies to Typhoon Kalmaegi.

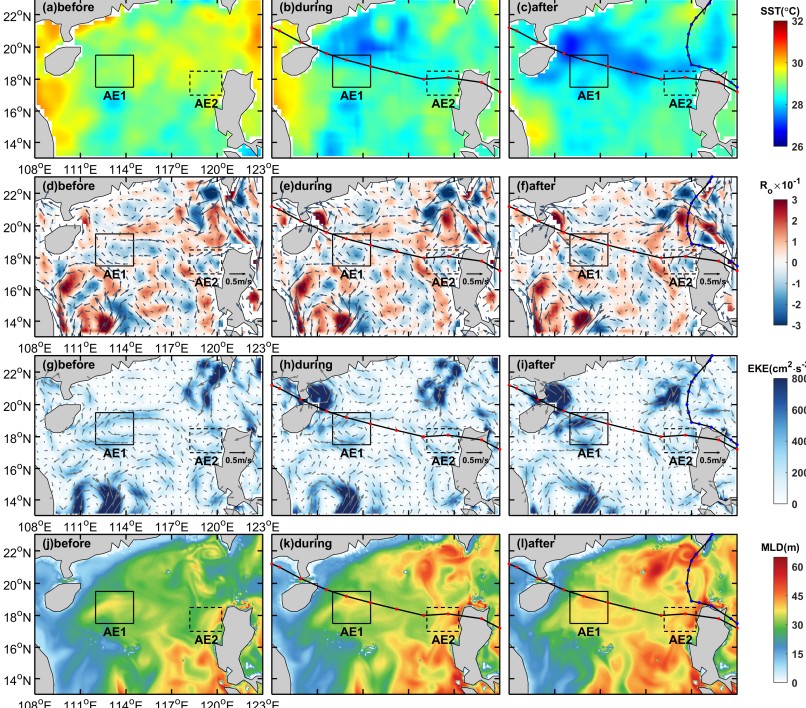


**Figure 5.** The spatial distribution of SST, $R_o$, EKE, and MLD before, during and after the passage of typhoon

Kalmaegi. The time periods of 10-13, 15-16 and 19-22 September are designated as stages before, during and after

typhoon, respectively. The path of typhoon Kalmaegi is depicted by a black solid line with red dots, while the path

of tropical storm Fung-wong is represented by a black solid line with blue dots in the third column. The solid and

dashed boxes correspond to AE1 and AE2, respectively.

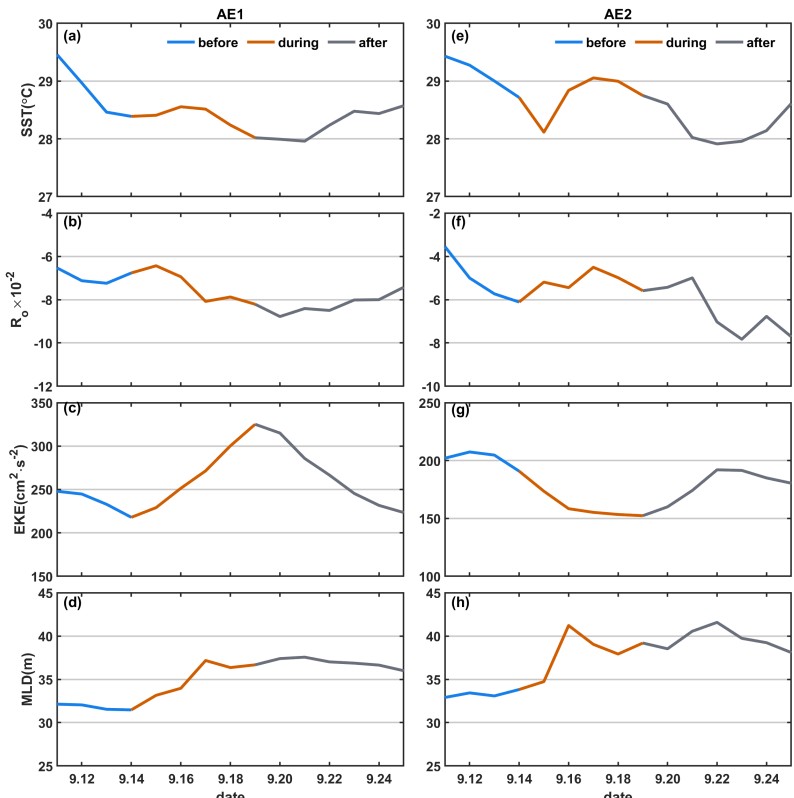

**Figure 6.** The time series of sea surface temperature (SST), $R_o$, eddy kinetic energy, and mixed layer depth (MLD) within the warm eddies' regions (black solid and dashed boxes in Fig. 5). The first coloum is variables of AE1, the second column is AE2.

**3.2 Upper-ocean vertical thermal and salinity structure of eddies**

We conducted further analysis on the vertical temperature and salinity structure of the warm eddies AE1 and AE2 before and after the typhoon Kalmaegi using GLORYS12V1 data. Fig. 7 illustrates that during the typhoon's passage on 15 September, the temperature above the MLD within AE1 increased by approximately 0.1 °C, while the salinity decreased by 0.02psu. Below the MLD, the temperature showed a significant increase, reaching a maximum temperature rise of 1.3 °C. Correspondingly, the salinity below the MLD exhibited a decrease of 0.05 psu. These changes led to a deepening of the isodensity by 15 m and a decrease in buoyancy frequency $N^2$ (Fig. 8a-b), indicating convergence and downwelling within the centre of the warm eddy AE1 (Fig. 4c-d).

After 15 September, the temperature above the MLD decreased and the salinity show an increase (Fig. 7a-b), resulting in the uplift of the 1021 kg m$^{-3}$ isodensity to the sea surface (Fig. 8a-b). The subsurface warming and salinity reduction gradually weakened after the typhoon Kalmaegi but persisted for about a week after the typhoon's passage until 22 September. This persistence can be attributed to the intensified stratification around MLD, with $N^2$ around $9.0 \times 10^{-4}$s$^{-2}$ (Fig. 8b). The increased stability





inhibits vertical mixing, restrains the exchange of heat and salinity, and leads to smoother density
gradients above the MLD (Fig. 8a).

The vertical temperature and salinity structure of AE2 exhibit an opposite trend. During the typhoon

passage on 15 September, AE2 also experienced a cooling trend of 0.2 °C, with a decrease in salinity of
0.04psu above the MLD. Below the MLD, the temperature showed a consistent decrease, with a change
of less than 0.5 °C within the subsurface. Correspondingly, the salinity exhibited an increase of
approximately 0.08 psu (Fig. 7c-d). The slightly upward shift of the isodensity (Fig. 8c) suggests the
possibility of cold-water upwelling induced by the suction effect of the typhoon. The temperature
decrease and salinity increase below the MLD were primarily driven by upwelling processes.

Furthermore, when the tropical storm Fung-wong passed through AE2 on 19 September (dashed line

in Fig. 7c-d), the decreasing trend of subsurface temperature became more pronounced, and the
subsurface salinity exhibited a significant increase. AE2 was more significantly influenced by the
typhoon Fung-wong. This can be attributed to the presence of a stable stratification with $N^2$ around
$8.4 \times 10^{-4} s^{-2}$ at a depth of 42 m, which created a barrier layer preventing the intrusion of high-salinity cold
water from the lower layers into the mixed layer (Yan et al., 2017).

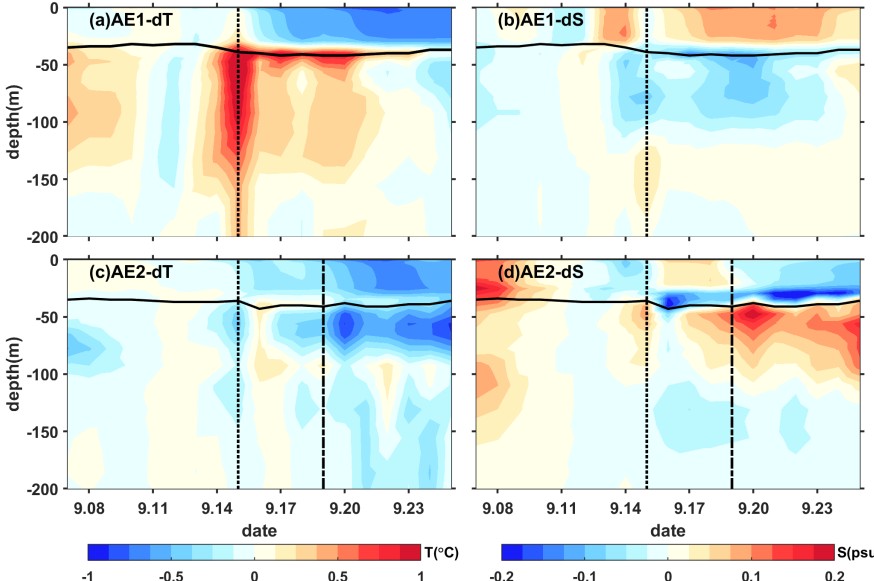

**Figure 7.** The timeseries of vertical temperature and salinity anomalies in the center of the warm eddies. The
anomalies were calculated relative to the average value of 10-13 September. The vertical black dotted line
indicates the typhoon Kalmaegi's passage, while the vertical black dashed line represents the passage of tropical
storm Fung-wong. The black solid line is the MLD.

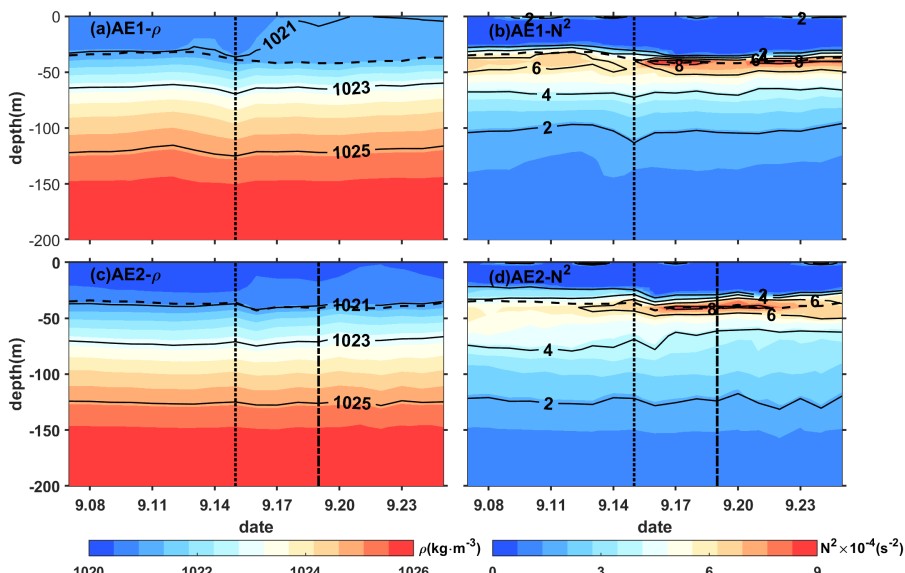

**Figure 8.** Same as Fig. 7, but for density and buoyancy frequency ($N^2$).

**3.3 Comparison of the response between eddies and non-eddies areas**

To investigate the contrasting response of warm eddies and the non-eddies background to typhoon Kalmaegi, we conducted a comparative analysis of vertical temperature and salinity profiles with these two areas. We examined data from Argo 2901469, which was located within AE2 during the period 11-19 September, while the temperature and salinity data from Station S5 was considered as the background, the S5 had a distance of 246 km from AE2's center on 15 September (Fig. 4). These profiles were categorized into three periods: pre-typhoon (11 September), during-typhoon (15 September), and post-typhoon (19 September).

At above 40m depth, both inside and outside of AE2 experienced a decrease in temperature, with a cooling of less than -1.0℃. Four days after the typhoon passage (19 September), the cooling persisted inside and outside the eddy, with the cooling being more pronounced outside the AE2, showing a decrease of 1.2 ℃ (Fig. 9c). The salinity within AE2 initially increased by 0.15 psu from the pre-typhoon stage to the during-typhoon stage and then decreased by 0.09 psu after the typhoon passage (Fig. 9d). In contrast, the salinity at Station S5 showed a similar pattern on pre-typhoon and during-typhoon stage, but increased by 0.05 psu after the typhoon. Two possible processes can explain the difference in salinity trends. First, during the pre-typhoon to typhoon stage, the entrainment within AE2 may have brought the subsurface water, which is saltier, up to the surface, resulting in an increase in salinity. The second process is related to the typhoon-induced precipitation after the typhoon passage, which led to a decrease in salinity. Strong stratification could have contributed to the persistence of saltier subsurface water. While in the S5, the increase in salinity was relatively minor only increased slightly.





On 15 September, the subsurface layer at 45 m to 100 m was affected by the cold upwelling caused
by the typhoon, resulting in a cooling and increased salinity within the AE2 warm eddy. As the typhoon's
forcing diminished, the upper layer of seawater began to mix, and influenced by the downward flow of
the eddy itself, warm surface water was transported to the subsurface layer. Four days later, a warming
phenomenon occurred, with the maximum warm anomaly of 1.2 °C observed at a depth of 75 m (Fig.
9a). The mixing effect outside the eddy was not significant, resulting in a slight subsurface warming of
approximately 0.2 °C, with no significant changes in salinity. However, on 19 September, a cooling
center of -1.2°C was observed at a depth of 60 m, corresponding to the maximum salinity anomaly of
0.13 psu (Fig. 9c-d). Below 100 m, the AE2 warm eddy experienced a temperature increase of 0.5 °C
and a slight decrease in salinity of 0.04 psu. On 19 September, the temperature and salinity within the
AE2 eddy showed little change. However, outside the eddy, a different response was observed. On
September 19th, a cooling trend was observed throughout the water column, within a range of 0.2 °C,
accompanied by a noticeable increase in salinity (Fig. 9c, d), within a range of 0.06 psu. This indicates
that the typhoon caused a significant upwelling outside the eddy region.

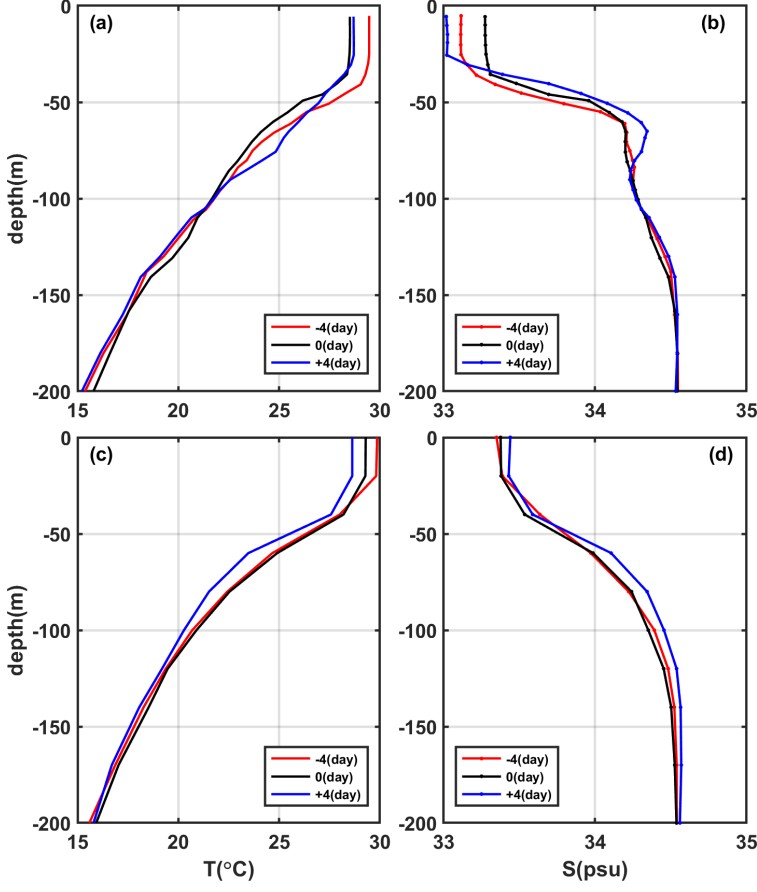




**Figure 9. (a-b)** the vertical profile of temperature and salt inside the eddy (Argo 2901469), **(c-d)** the vertical profiles
of temperature and salt outside the eddy (S5). The red, black and blue lines represent pre-typhoon, during-typhoon
and post-typhoon stages.

Based on Argo profiles and S5 data, the upper ocean above 200 m inside and outside the eddy

responded differently to the forcing of the typhoon. In the upper layer (0-40m), cooling was observed
both inside and outside the eddy, and it lasted for a longer duration. In the subsurface layer (45-100m),
after the passage of the typhoon (19 September), there was a strong cooling outside the eddy, while
warming occurred within the warm eddy AE2. Zhang (2022) pointed out that the sea temperature
anomalies mainly depend on the combined effects of mixing and vertical advection (cold suction).
Mixing causes surface cooling and subsurface warming, while upwelling (downwelling) leads to cooling
(warming) of the entire upper ocean. The temperature anomaly in the subsurface layer depends on the
relative strength of mixing and vertical advection, with cold anomalies dominating when upwelling is
strong, and downwelling amplifying the warming anomalies caused by mixing. Therefore, due to the
strong influence of upwelling outside the eddy, the temperature profile of the entire water column shifts
upwards, resulting in cooling of the entire upper ocean. On the other hand, influenced by the downwelling
associated with the warm eddy itself, a warming anomaly of 1.2 °C is observed in the subsurface layer.
Compared to region AE2, the cold suction effect caused by the typhoon Kalmaegi is still evident in the
non-eddy area.
**4. Discussion**

From the above, the relative position of warm eddies and the typhoon can influence the response of

the eddies(Lu et al., 2020). The warm eddy AE1, located on the left side of the typhoon track, was not
weakened by the strong cold suction effect caused by the typhoon Kalmaegi. Instead, it was strengthened
due to the stronger negative wind stress curl generated by the typhoon. Starting from 15 September, there
was a significant positive sea level anomaly (SLA) to the west of 113.5°E, and its intensity increased,
reaching its maximum on 20 September (Fig. 10a). This strengthening is consistent with the increase in
the amplitude of the warm core of the eddy AE1. Comparing with the wind stress curl anomaly (Fig.
10b), it can be seen that from 15 to 16 September, the typhoon Kalmaegi moved over the section at
18.2°N, specifically to the west of 113.5°E, exhibited strong negative wind stress curl anomalies, with a
maximum intensity of $-3 \times 10^{-6}$ N.m$^{-3}$. The negative wind stress curl induced by the typhoon resulted in
favourable surface ocean currents that further enhanced the anticyclonic spin of the warm eddy. The
negative wind stress curl anomaly caused strong downwelling currents, inputting negative vorticity into
AE1, leading to its intensification (Fig. 6b-c), as indicated by the enhanced positive SLA (Fig. 10a).
Conversely, the region to the east of 113.5°E along the section exhibited negative SLA anomalies. This
weakening is consistent with the previous observations of the intensified warm core and decreased eddy
area in the eddy AE1.

Comparing with the wind stress curl anomaly (Fig. 10b), it can be seen that from 15-16 September,

there is a strong positive wind stress curl anomaly at the center section of AE1, specifically at 114°E,





with a maximum intensity of $3\times10^{-6}$ N m$^{-3}$. The positive wind stress curl induces upwelling, which inputs
the positive vorticity of the typhoon-induced wind stress curl downward into the eddy (Huang and Wang,
2022), corresponding to the decrease in SLA (Fig. 10a).

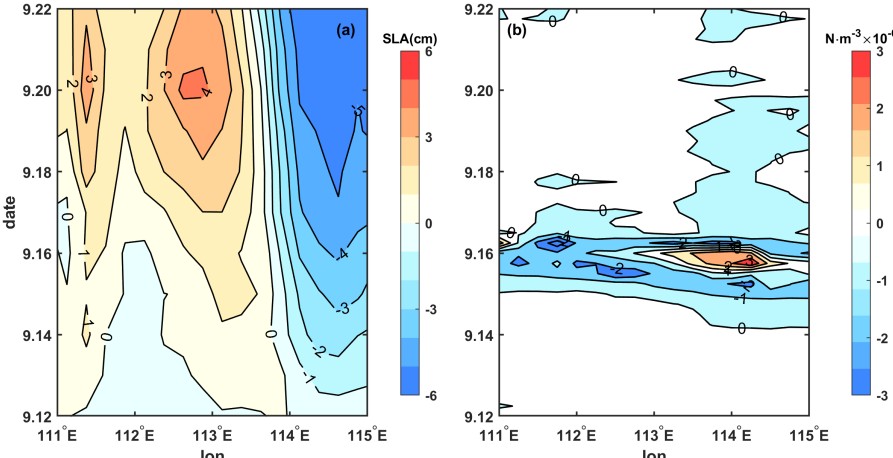


**Figure 10.** The time/longitude plots of **(a)** SLA anomaly (cm) and **(b)** wind stress curl (N.m$^{-3}$) anomaly at the central
section of AE1 (18.2 °N). The anomalies were calculated relative to the average value of 10-13 September.
The response of the warm eddy AE2 is different from AE1 mainly because AE2 is quite near the
typhoon track, and the significantly positive wind stress curl at the center of the typhoon noticeably
weakens the eddy. Furthermore, based on the meridional isotherm profiles of the eddy center at three
periods, it can be observed that during the passage of Typhoon Kalmaegi (15 September), the isotherms
in the AE1 region exhibit significant subsidence (Fig. 11a), while in the AE2 region, the isotherms show
uplift (Fig. 11b). This result is consistent with the earlier finding that the convergence and subsidence
within the warm eddy AE1 are enhanced by the influence of the wind stress curl induced by the typhoon,
while the intensity of AE2 is weakened.
To understand the work done by the typhoon on the eddy in the ocean, we estimate the total work
inputted into the ocean current $u_c$ using the previously calculated wind stress (Liu et al., 2017):
$$W = \int \vec{\tau} \cdot \overrightarrow{u_c} \, dt \ . \tag{10}$$
Here, we select the region near the typhoon track where the wind speed is greater than 17 m.s$^{-1}$ as the
typhoon forcing region to understand the energy inputted by the typhoon to the warm eddy (Sun et al.,
2010). The forcing duration over the ocean in the typhoon-affected region and the work done by the
typhoon on the surface current are shown in Fig. 12. When the angle between the wind and the ocean
current is acute, the typhoon does positive work on the ocean current. Conversely, when the angle is
obtuse, the typhoon does negative work on the ocean current. It can be observed that the region with the
maximum forcing duration by the typhoon on AE1 is also the area where the typhoon clearly does
positive work on the ocean current, with a cumulative work done exceeding 8 KJ m$^{-2}$. This accelerates
the flow velocity in the eddy, resulting in convergence within the eddy and an increase in SLA, leading
to the strengthening of AE1. On the other hand, the forcing duration by the typhoon on AE2 is smaller,





and the typhoon does negative work on the ocean current in most areas, with a cumulative work done
within -5 KJ m⁻², causing the flow velocity within the AE2 to decelerate. The center height decreases and
AE2 weakens.

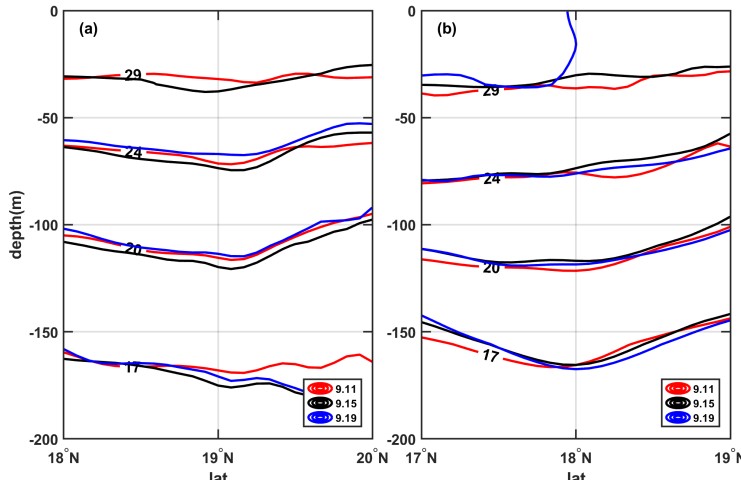

**Figure 11.** The meridional isotherm profiles of AE1 **(a)** and AE2 **(b)** before (11 September), during (15 September)
and after (19 September) typhoon Kalmaegi,

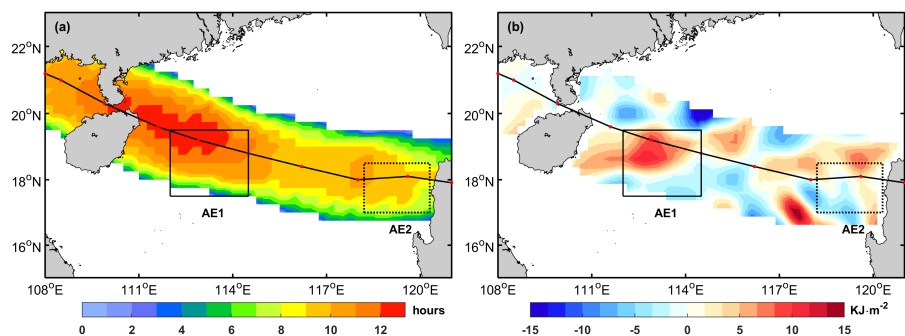

**Figure 12.** **(a)**: the forcing time (unit: hours) of the typhoon; **(b)**: the input work (unit: KJ. m⁻²) of the typhoon to
the current.
**5. Summary**

Based on multi-satellite observations, on-site measurements, and numerical model data, we have

gained valuable insights into the response of warm eddies AE1 and AE2 in the northern South China Sea
to Typhoon Kalmaegi. Both horizontally and vertically, these eddies displayed distinct differences.
Horizontally, we observed a reduction in their respective areas by approximately 31% (AE1) and 36%
(AE2). AE1, positioned on the left side of the typhoon's track, strengthened with amplitude, $R_O$ and EKE
increasing by 1.3 cm, $1.4\times10^{-2}$ and 107.2 cm² s⁻² after the typhoon passed. In contrast, AE2, which
intersected with the typhoon's track, weakened with amplitude, $R_O$ and EKE decreasing by 3.1 cm,



$1.6 \times 10^{-2}$ and 38.5 cm$^2$ s$^{-2}$, respectively. Vertically, during the typhoon's passage, AE1 experienced
intensified converging subsidence flow at its center, leading to an increase in temperature and a decrease
in salinity above depth of 150m. This response was more pronounced below the MLD (1.3°C) and
persisted for about a week after the typhoon. On the other hand, AE2 exhibited cooling above the MLD,
accompanied by a decrease in salinity, as well as a subsurface temperature drop and salinity increase due
to the upwelling of cold water caused by the typhoon's suction effect. The subsurface cooling and salinity
increase in AE2 were further influenced by Typhoon Fung-wong. Additionally, from the temperature
vertical profile of Argo and in-situ arrays, on 19 September, it can be seen that the non-eddy region also
experienced significant cooling, with a prominent cooling center observed at a depth of 60 m (-1.2 °C).
The warm eddy AE2, influenced by its own downwelling, exhibited enhanced mixing effects, resulting
in a subsurface warm anomaly of 1.2 °C.

Further analysis reveals that the different responses of the warm eddies can be attributed to factors
such as wind stress curl distribution, which are influenced by the relative position of the warm eddies
and the typhoon track. The wind stress curl induced by the typhoon plays a crucial role in shaping the
response of the warm eddies. AE1, located on the left side of the typhoon's path, experienced prolonged
forcing from the typhoon, resulting in positive work on the ocean current. This inputted a strong negative
wind stress curl into the eddy, enhancing negative EPV, so the downwelling within the AE1 is obvious
and contributing to its increased strength. In contrast, AE2, positioned directly below the typhoon's track,
experienced shorter forcing duration and weakened due to the strong positive wind stress curl at the
typhoon's center. Furthermore, the absolute value of EPV increased in both warm eddies during the
typhoon's passage, but with differing impacts. The positive EPV contributed to surface water cooling and
the influx of cooler subsurface water, while the negative EPV facilitated downwelling and intensified the
influence of the warm eddies.

In summary, the different responses of warm eddies to typhoons provide valuable insights into the
complex interactions between the atmosphere and the ocean. Understanding these responses is crucial
for accurate climate modeling and weather forecasting. By investigating factors such as wind stress curl
distribution, EPV, buoyancy frequency and the relative position of the eddies to the typhoon's track,
researchers can gain a more precise understanding of the underlying mechanisms driving these
interactions. This knowledge contributes to improved predictions and mitigation strategies for the
impacts of typhoons and other extreme weather events, enhances the accuracy of climate models, and
advances weather forecasting capabilities.



*Data av*ailability. The six-hourly best-track typhoon datasets were accessed on 3 February 2021 by JTWC,
http://www.usno.navy.mil/JTWC,    JMA,    https://www.jma.go.jp/jma/jma-eng/jma-center/rsmc-hp-pub-
eg/besttrack.html and CMA, http://tcdata.typhoon.gov.cn. The AVISO product was accessed on 14 February
2021 by https://marine.copernicus.eu/. The AVHRR SST data was accessed on 16 March, 2022 by
ftp://podaac.jpl.nasa.gov/documents/dataset_docs/avhrr_pathfinder_sst.html. The Argo data was accessed
on 4 April, 2022 by https://dataselection.euro-argo.eu/. The wind data was accessed on 5 January, 2023 by
https://apps.ecmwf.int/datasets/data/interim-full-daily/levtype=sfc/. The GLORYS12V1 was accessed on
23 March, 2022 by https://marine.copernicus.eu/.
*Author contributions.* XYL and HZ contributed to the study conception and design. Material preparation, data
collection and analysis were performed by YHH and XYL. GQH and YL contributed to the methodology. The
original manuscript was prepared by XYL and YHH. All the authors contributed to the review and editing of
the manuscript.
*Competing interests.* The contact author has declared that none of the authors has any competing interests.
*Disclaimer.* Publisher's note: Copernicus Publications remains neutral with regard to jurisdictional claims in
published maps and institutional affiliations.
*Acknowledgements.* These data were collected and made freely available by JTWC, JMA, CMA, AVISO, AVHRR,
Argo, ECMWF, COPERNICUS. All figures were created using MATLAB, in particular using the M_Map toolbox
(Pawlowicz, 2020). The authors thank the anonymous reviewers, whose feedback led to substantial im-
provement of the resulting analyses, figures and manuscript
*Financial support.* This research has been supported by the National Natural Science Foundation of China
(42227901), Southern Marine Science and Engineering Guangdong Laboratory (Zhuhai), grant number
SML2020SP007 and SML2021SP207; the Innovation Group Project of Southern Marine Science and
Engineering Guangdong Laboratory (Zhuhai), grant number 311020004 and 311022001; the National
Natural Science Foundation of China, grant number 42206005; the open fund of State Key Laboratory of
Satellite Ocean Environment Dynamics, Second Institute of Oceanography, MNR, grant number QNHX2309;
General scientific research project of Zhejiang Provincial Department of Education, grant number
Y202250609; the Open Foundation from Marine Sciences in the First-Class Subjects of Zhejiang, grant number
OFMS006; State Key Laboratory of Tropical Oceanography (South China Sea Institute of Oceanology Chinese
Academy of Sciences), grant number LTO2220.





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

A Preliminary Study from Satellite Observations and Numerical Modelling, TAO: Terrestrial,
Atmospheric and Oceanic Sciences, 22,https://doi.org/10.3319/tao.2011.08.19.01(tm), 2011.
Lin, I. I., Wu, C.-C., Emanuel, K. A., Lee, I. H., Wu, C.-R., and Pun, I.-F.: The Interaction of
Supertyphoon Maemi (2003) with a Warm Ocean Eddy, Mon. Weather Rev., 133, 2635-
2649,https://doi.org/10.1175/MWR3005.1, 2005.
Liu, F. and Tang, S.: Influence of the Interaction Between Typhoons and Oceanic Mesoscale Eddies on
Phytoplankton Blooms, J. Geophys. Res.: Oceans, 123, 2785-
2794,https://doi.org/10.1029/2017jc013225, 2018.



Liu, S.-S., Sun, L., Wu, Q., and Yang, Y.-J.: The responses of cyclonic and anticyclonic eddies to
typhoon forcing: The vertical temperature-salinity structure changes associated with the horizontal
convergence/divergence,    J.    Geophys.    Res.:    Oceans,    122,    4974-
4989,https://doi.org/10.1002/2017JC012814, 2017.
Lu, Z., Wang, G., and Shang, X.: Response of a Preexisting Cyclonic Ocean Eddy to a Typhoon, J. Phys.
Oceanogr., 46, 2403-2410,https://doi.org/10.1175/jpo-d-16-0040.1, 2016.
Lu, Z., Wang, G., and Shang, X.: Strength and Spatial Structure of the Perturbation Induced by a Tropical
Cyclone to the Underlying Eddies, J. Geophys. Res.: Oceans, 125,https://doi.org/10.1029/2020jc016097,
614  2020.

Lu, Z., Wang, G., and Shang, X.: Observable large-scale impacts of tropical cyclones on subtropical gyre,
J. Phys. Oceanogr.,https://doi.org/10.1175/JPO-D-22-0230.1, 2023.
Ma, Z., Zhang, Z., Fei, J., and Wang, H.: Imprints of Tropical Cyclones on Structural Characteristics of
Mesoscale    Oceanic    Eddies    Over    the    Western    North    Pacific,    Geophys.    Res.    Lett.,
48,https://doi.org/10.1029/2021gl092601, 2021.
Ma, Z., Fei, J., Liu, L., Huang, X., and Li, Y.: An Investigation of the Influences of Mesoscale Ocean
Eddies    on    Tropical    Cyclone    Intensities,    Mon.    Weather    Rev.,    145,    1181-
1201,https://doi.org/10.1175/mwr-d-16-0253.1, 2017.
Mei, W., Lien, C.-C., Lin, I. I., and Xie, S.-P.: Tropical Cyclone–Induced Ocean Response: A
Comparative Study of the South China Sea and Tropical Northwest Pacific*,+, J. Clim., 28, 5952-
5968,https://doi.org/10.1175/jcli-d-14-00651.1, 2015.
Mitarai, S. and McWilliams, J. C.: Wave glider observations of surface winds and currents in the core of
Typhoon Danas, Geophys. Res. Lett., 43, 11312-11319,https://doi.org/10.1002/2016gl071115, 2016.
Ning, J., Xu, Q., Zhang, H., Wang, T., and Fan, K.: Impact of Cyclonic Ocean Eddies on Upper Ocean
Thermodynamic Response to Typhoon Soudelor, Remote Sens., 11,https://doi.org/10.3390/rs11080938,
630  2019.

Oey, L. Y., Ezer, T., Wang, D. P., Fan, S. J., and Yin, X. Q.: Loop Current warming by Hurricane Wilma,
Geophys. Res. Lett., 33,https://doi.org/10.1029/2006gl025873, 2006.
Price, J. F.: Upper Ocean Response to a Hurricane, J. Phys. Oceanogr.,https://doi.org/10.1175/1520-
0485(1981)011%3C0153:UORTAH%3E2.0.CO;2, 1981.
Pujol, M.-I., Faugère, Y., Taburet, G., Dupuy, S., Pelloquin, C., Ablain, M., and Picot, N.: DUACS
DT2014: the new multi-mission altimeter data set reprocessed over 20 years, Ocean Sci., 12, 1067-
1090,https://doi.org/10.5194/os-12-1067-2016, 2016.
Rudzin, J. E. and Chen, S.: On the dynamics of the eradication of a warm core mesoscale eddy after the
passage    of    Hurricane    Irma    (2017),    Dyn.    Atmos.    Oceans,
100,https://doi.org/10.1016/j.dynatmoce.2022.101334, 2022.
Shang, X.-d., Zhu, H.-b., Chen, G.-y., Xu, C., and Yang, Q.: Research on Cold Core Eddy Change and
Phytoplankton Bloom Induced by Typhoons: Case Studies in the South China Sea, Adv. Meteorol., 2015,
1-19,https://doi.org/10.1155/2015/340432, 2015.
Shay, L. K. and Jaimes, B.: Mixed Layer Cooling in Mesoscale Oceanic Eddies during Hurricanes
Katrina and Rita, Mon. Weather Rev., 137, 4188-4207,https://doi.org/10.1175/2009mwr2849.1, 2009.
Shay, L. K. and Jaimes, B.: Near-Inertial Wave Wake of Hurricanes Katrina and Rita over Mesoscale
Oceanic Eddies, J. Phys. Oceanogr., 40, 1320-1337,https://doi.org/10.1175/2010jpo4309.1, 2010.





Shay, L. K., Goni, G. J., and Black, P. G.: Effects of a Warm Oceanic Feature on Hurricane Opal, Mon.
Weather Rev., 128, 1366-1383,https://doi.org/10.1175/1520-
0493(2000)128<1366:EOAWOF>2.0.CO;2, 2000.
Song, D., Guo, L., Duan, Z., and Xiang, L.: Impact of Major Typhoons in 2016 on Sea Surface Features
in the Northwestern Pacific, Water, 10,https://doi.org/10.3390/w10101326, 2018.
Stramma, L., Cornillon, P., and Price, J. F.: Satellite observations of sea surface cooling by hurricanes,
J. Geophys. Res.: Oceans, 91, 5031-5035,https://doi.org/10.1029/JC091iC04p05031, 1986.
Sun, J., Ju, X., Zheng, Q., Wang, G., Li, L., and Xiong, X.: Numerical Study of the Response of Typhoon
Hato (2017) to Grouped Mesoscale Eddies in the Northern South China Sea, J. Geophys. Res.: Atmos.,
128,https://doi.org/10.1029/2022jd037266, 2023.
Sun, L., Yang, Y., Xian, T., Lu, Z., and Fu, Y.: Strong enhancement of chlorophyll a concentration by a
weak typhoon, Mar. Ecol. Prog. Ser., 404, 39-50,https://doi.org/10.3354/meps08477, 2010.
Sun, L., Li, Y.-X., Yang, Y.-J., Wu, Q., Chen, X.-T., Li, Q.-Y., Li, Y.-B., and Xian, T.: Effects of super
typhoons on cyclonic ocean eddies in the western North Pacific: A satellite data-based evaluation
between 2000 and 2008, J. Geophys. Res.: Oceans, 119, 5585-
5598,https://doi.org/10.1002/2013jc009575, 2014.
Thompson, B. and Tkalich, P.: Mixed layer thermodynamics of the Southern South China Sea, Clim.
Dyn., 43, 2061-2075,https://doi.org/10.1007/s00382-013-2030-3, 2014.
Vincent, E. M., Lengaigne, M., Madec, G., Vialard, J., Samson, G., Jourdain, N. C., Menkes, C. E., and
Jullien, S.: Processes setting the characteristics of sea surface cooling induced by tropical cyclones, J.
Geophys. Res.: Oceans, 117, n/a-n/a,https://doi.org/10.1029/2011JC007396, 2012.
Wada, A. and Usui, N.: Impacts of Oceanic Preexisting Conditions on Predictions of Typhoon Hai-Tang
in 2005, Adv. Meteorol., 2010, 756071,https://doi.org/10.1155/2010/756071, 2010.
Walker, N. D., Leben, R. R., and Balasubramanian, S.: Hurricane-forced upwelling and
chlorophyllaenhancement within cold-core cyclones in the Gulf of Mexico, Geophys. Res. Lett., 32, n/a-
n/a,https://doi.org/10.1029/2005gl023716, 2005.
Wang, G., Su, J., Ding, Y., and Chen, D.: Tropical cyclone genesis over the south China sea, J. Mar.
Syst., 68, 318-326,https://doi.org/10.1016/j.jmarsys.2006.12.002, 2007.
Wang, G., Zhao, B., Qiao, F., and Zhao, C.: Rapid intensification of Super Typhoon Haiyan: the
important role of a warm-core ocean eddy, Ocean Dyn., 68, 1649-1661,https://doi.org/10.1007/s10236-
018-1217-x, 2018.
Wu, C.-R., Chiang, T.-L., and Oey, L.-Y.: Typhoon Kai-Tak: An Ocean's Perfect Storm, J. Phys.
Oceanogr., 41, 221-233,https://doi.org/10.1175/2010JPO4518.1, 2011.
Xiu, P., Chai, F., Shi, L., Xue, H., and Chao, Y.: A census of eddy activities in the South China Sea
during 1993–2007, J. Geophys. Res.: Oceans, 115,https://doi.org/10.1029/2009jc005657, 2010.
Yan, Y., Li, L., and Wang, C.: The effects of oceanic barrier layer on the upper ocean response to tropical
cyclones, J. Geophys. Res.: Oceans, 122, 4829-4844,https://doi.org/10.1002/2017jc012694, 2017.
Yu, F., Yang, Q., Chen, G., and Li, Q.: The response of cyclonic eddies to typhoons based on satellite
remote sensing data for 2001–2014 from the South China Sea, Oceanologia, 61, 265-
275,https://doi.org/10.1016/j.oceano.2018.11.005, 2019.
Yu, J., Lin, S., Jiang, Y., and Wang, Y.: Modulation of Typhoon-Induced Sea Surface Cooling by
Preexisting Eddies in the South China Sea, Water, 13,https://doi.org/10.3390/w13050653, 2021.



Zhang, H.: Modulation of Upper Ocean Vertical Temperature Structure and Heat Content by a Fast-
Moving Tropical Cyclone, J. Phys. Oceanogr., 53, 493-508,https://doi.org/10.1175/jpo-d-22-0132.1,
692  2022.
Zhang, H., Chen, D., Zhou, L., Liu, X., Ding, T., and Zhou, B.: Upper ocean response to typhoon
Kalmaegi (2014), J. Geophys. Res.: Oceans, 121, 6520-6535,https://doi.org/10.1002/2016jc012064,
695  2016.
Zhang, Y., Zhang, Z., Chen, D., Qiu, B., and Wang, W.: Strengthening of the Kuroshio current by
intensifying tropical cyclones, Science, 368, 988-993,https://doi.org/10.1126/science.aax5758, 2020.