# Peer review of "The Different Dynamic Influences of Typhoon Kalmaegi on two Pre-existing Anticyclonic Ocean Eddy"

_EGUsphere, 2023_

## Referee Comment (RC1)

I don't think this paper is publishable; so sorry.

General Comments: Kalmaegi was a fast-moving TC. At 8 m/s, the TC traverses ~600 km in 1 day and spent ~6 hours traversing AE1 (or AE2) with a diameter of about 150 km. In such a super-critically moving storm, most of the wind effect on the ocean is therefore through mixing (including perhaps that caused by near-inertial internal wave breaking in the upper ocean in the wake of the storm) rather than the wind stress curl. The latter would require that wind acts on the ocean in a time scale longer than the inertial period (~1.5 days at 19N). I understand the authors' hypothesis of the negative WSC (thus convergence) on the left side, etc., but I don't think it is a demonstrable one in this case and is most likely incorrect. The increased AE1 after Kalmaegi (Fig.3, etc.) is likely a complex eddy adjustment process. One may suspect such adjustment also from Fig.3 in which the "warm" area between AE1 and AE2, including that on the left side, shrinks or weakens. That area would have expanded following the authors' hypothesis.

Two other general comments. 1) AE1's increased amplitude, Ro and EKE = 1.3 cm, 1.4e-2 and 107 (cm/s)^2 are small. Are they statistically significant, and were errors and confidence levels estimated? Similarly for AE2. 2) Inertial oscillatory response persists long (~5 days and longer) after a storm passes (see e.g. Wu et al. Effect of Typhoon Kalmaegi (2014) on northern South China Sea explored using Muti-platform satellite and buoy observations data; Prog Oceanogr. 180 (2020) 102218). The effects of inertial motions on the Authors' results and analyses were not discussed and I am unsure, for example, how the effects were filtered out or accounted for and how they may affect their estimates.

Other Comments:
L14: Rossby number (Ro = relative vorticity/Coriolis parameter);
L16: Rossby number;
L166: vertical feedback of the ocean by ... Kalmaegi: Not sure what this means, what "feedback", maybe "response..."?
Also: I assume GLORY assimilates Argo data but not the Station data. If so, then it is unsurprising that GLORY agrees with ARGO but not Station 5 (Figure 1).
Only Station 5 on the left side of the storm was used to support the authors' hypothesis. To support (refute?) the Authors' hypothesis I suggest using data from other Stations (except #3), right and left of Kalmaegi.
L223: ... weak wind stress curl, to be more precise. The term "wind shear" is also customarily taken as "vertical wind shear" in TC studies in meteorology so can be confusing.
L245: ... with 6-hourly dots.
 .
 .

---

## Author Comment (AC1)

**Response to Reviewer 1**

General Comments: Kalmaegi was a fast-moving TC. At 8 m/s, the TC traverses ~600 km in 1 day and spent ~6 hours traversing AE1 (or AE2) with a diameter of about 150 km. In such a super-critically moving storm, most of the wind effect on the ocean is therefore through mixing (including perhaps that caused by near-inertial internal wave breaking in the upper ocean in the wake of the storm) rather than the wind stress curl. The latter would require that wind acts on the ocean in a time scale longer than the inertial period (~1.5 days at 19N). I understand the authors' hypothesis of the negative WSC (thus convergence) on the left side, etc., but I don't think it is a demonstrable one in this case and is most likely incorrect. The increased AE1 after Kalmaegi (Fig.3, etc.) is likely a complex eddy adjustment process. One may suspect such adjustment also from Fig.3 in which the "warm" area between AE1 and AE2, including that on the left side, shrinks or weakens. That area would have expanded following the authors' hypothesis.

**Response: We would like to thank you for your careful reading, helpful comments, and constructive suggestions, which have significantly improved the presentation of our manuscript. We have carefully considered all comments from the reviewer and revised our manuscript accordingly. The manuscript has also been double-checked, the typos and grammar errors we found have been corrected. The changes are highlighted in the manuscript. All page numbers refer to the revised manuscript file with tracked changes.**

Thank you for your comments.

We agree with the reviewer that during the forcing stage of a fast typhoon, there are exsits near inertial waves and they are very important. Here we show the snapshots of AE1 through its lifetime from 26 June to 14 October, 2014. We can see that the entire buoy array located within the AE1 from 31 July to 14 August (Figure S1). It can explain the near-inertial waves propagated downward into ocean interior from this period (Figure S2). It can be seen that near-inertial currents during 18 July to 19 August and 16-30 September, with a maximum

near-inertial velocity of 0.61 cm s$^{-1}$, which are affected by typhoon Rammasun (16-18, July) and Kalmaegi (14-16, September). The near-inertial energy of Kalmaegi lasts from surface to 200 m, but near-inertial currents caused by typhoon Rammasun lasts longer, it stays at upper 50 m on 18-24 July, then the near-inertial energy enters and traps in AE1, it transmits downward since 25 July and stay at 50-200 m until 17 August. The transfer of energy from anticyclonic eddy to near-inertial waves is the main reason for the downward progpation and longtime perisistence of near-inertial energy (Chen et al, 2023). The near-inertial velocity distribution pattern of Station 4 during the period from 30 July to 19 August is different from station 1 and 4, because AE1 gradually moves away from station 4 (located in the northeast of the bouy array, Fig. S1), it captures weakest near-inertial energy. Due to the westward movement of AE1, the eddy center is near station 2, only station 2 catches the subsurface near-inertial signals during the period of 12 August to 8 September (the red box area in Fig. S2), and it is relatively small, with a maximum velocity of 0.18 cm s$^{-1}$.

[Figure]

**Figure S1.** Eddy boundarys of AE1 and its distance from the buoy array during its lifecycle (26 June to 8 October). The color and gray contour lines represent sea level anomaly, while the black solid contours are AE1's boundaries. The five red dots represent the positions of 5 buoys.

[Figure]

**Figure S2.** Eastward(a,c,e) and northward(b,d,f) near-inertial currents in the upper 200 m observed at station 1, 2 and 4.

Chen Z, Yu F, Chen Z, et al. Downward Propagation and Trapping of Near-Inertial Waves by a Westward-moving Anticyclonic Eddy in the Subtropical Northwestern Pacific Ocean[J]. Journal of Physical Oceanography, 2023.

Since our corresponding author Han Zhang has previously published 6 papers (listed at below) and discussed the ocean response of typhoon Kalmaegi from multiple perspectives, including variations of near-inertial energy, vertical temperature, heat changes and their mechanisms during typhoon, so in this paper, near-inertial effect and mixing is not the focus. Moreover, AE1 is already far from the buoy array during typhoon Kalmaegi passed over NSCS, so the near-inertial waves at this period has little impact on AE1 and is excited by the first baloclinic mode (Zhang et al, 2017). Furthermore, daily reanalysis data is insufficient to study near-inertial waves in AE1 at this time.

Zhang H. Modulation of upper ocean vertical temperature structure and heat content by a fast-moving tropical cyclone[J]. Journal of Physical Oceanography, 2023, 53(2): 493-508.

Hong W, Zhou L, Xie X, Zhang H, Liang C. Modified parameterization for near-inertial waves. Acta Oceanologica Sinica, 2022, 41(10): 41-53. https://doi.org/10.1007/s13131-022-2012-6

Zhang H, Wu R, Chen D, Liu X, He H, Tang Y, Ke D, Shen Z, Li J, Xie J, Tian D, Ming J, Liu F, Zhang D, Zhang W. Net Modulation of Upper Ocean Thermal Structure by Typhoon Kalmaegi (2014). Journal of Geophysical Research: Oceans, 2018, 123(10): 7154-7171.

Zhang H, Chen D, Zhou L, Liu X, Ding T, Zhou B. Upper ocean response to typhoon Kalmaegi (2014). Journal of Geophysical Research: Oceans, 2016, 121(8): 6520-6535.

Wu R, Zhang H, Chen D, et al. Impact of Typhoon Kalmaegi (2014) on the South China Sea: Simulations using a fully coupled atmosphere-ocean-wave model[J]. Ocean Modelling, 2018, 131: 132-151.

Wu R, Zhang H, Chen D. Effect of Typhoon Kalmaegi (2014) on northern South China Sea explored using Muti-platform satellite and buoy observations data[J]. Progress in Oceanography, 2020, 180: 102218.

In addition, we believe that the ocean responds quickly to wind stress curl caused by typhoon Kalmaegi with no time lag. Ekman layer depth ($D_E$) varied with typhoon passage is shown in Fig. S3, when Kalmaegi approaches at 0000 UTC on 14 September, the mean $D_E$ within AE1 is only 21 m, while AE2 is 114 m, indicates that AE2 has already influenced by typhoon Kalmaegi. Then $D_E$ of AE2 sharply deepens, reaching a maximum depth of 241 m (Fig. S4) at 0000 UTC on 15 September when the center of Kalmegi is near AE2. As Kalmaegi moved northwest, the $D_E$ within AE1 reached its maximum depth of 262 m at 0000UTC on 16 September. The trends of $D_E$ within AE1 and AE2 are almost consistent, but AE1 lags AE2 by one day. From 15 September, $D_E$ within AE1 and AE2 gradually shallower, with the minimum $D_E$ of 60 m, which is 28 m higher than before typhoon, indicating that typhoon's effect through wind is still exist. For AE2, $D_E$ reached its minimum of 45 m at 0000 UTC on September, later increased gradually under the influence of tropical storm Fung-wong.

Due to the fact that near-inertial oscillation mainly manifests as the transfer of vertical energy, and Ekman Pumping can truly bring about vertical velocity changes, we believe that the theory of Ekman Pumping can be used to explain the vertical variation of AE1 and AE2. We have added these sentences at lines 447-456.

[Figure]

**Figure S3.** Ekman layer depth (DE) from 14 September to 18 September (a-i). The color represents the DE, the blue solid line is the path of Kalmaegi, the red dot and diamond are the positions of Station 5 and Argo 2901469 on 15 September, respectively. (Fig. 10 in manuscript)

[Figure]

**Figure S4.** Timeseries of $D_E$ from 14 September to 18 September within AE1 (a) and AE2 (b), respectively. The red line is the maximum $D_E$ and the blue line represents the mean maximum.

Regarding the second question, it can be seen from Figure 9 (line 458) that during typhoon Kalmaegi, the Ekman Pumping Velocity (EPV) on the left side of the typhoon path has both positive and negative values, so there exists both upwelling and downwelling on the left side of the path. In AE1, vertical velocity is downwelling, and most other places on the left side are upwelling. With the effect of advection, the overall cooling effect is greater than the warming effect, so the warm area is decreasing.

Two other general comments. 1) AE1's increased amplitude, Ro and EKE = 1.3 cm, 1.4e-2 and 107 (cm/s)^2 are small. Are they statistically significant, and were errors and confidence levels estimated? Similarly for AE2. 2) Inertial oscillatory response persists long (~5 days and longer) after a storm passes (see e.g. Wu et al. Effect of Typhoon Kalmaegi (2014) on northern South China Sea explored using Muti-platform satellite and buoy observations data; Prog Oceanogr. 180 (2020) 102218). The effects of inertial motions on the Authors' results and analyses were not discussed and I am unsure, for example, how the effects were filtered out or accounted for and how they may affect their estimates.

**Response:** Thank you. Because there are only two eddies studied here, too few samples to conduct significant testing. Although the increase (decrease) of amplitude, $R_o$, EKE of AE1 (AE2) are small, their proportion is not small compared to their average value. So we add these sentences on abstract at lines 14-17:

The amplitude, Rossby number (Ro) and eddy kinetic energy (EKE) of AE1 increased by 1.3 cm (5.7%), $1.4\times10^{-2}$ (20.6%) and 107.2 cm2 s$^{-2}$ (49.2%) after the typhoon, respectively, while AE2 weakened and the amplitude, vorticity and EKE decreased by 3.1 cm (14.6%), $1.6\times10^{-2}$ (26.2%) and 38.5 cm2 s$^{-2}$ (20.2%), respectively.

2) Thank you for recommending a very good paper and results. We have cited some of the conclusions from Wu et al (2020) at introduction of line xx. From Fig. S2, the near-inertial oscillation can persistence longer than 1 month.

Wu R, Zhang H, Chen D, et al. Impact of Typhoon Kalmaegi (2014) on the South China Sea: Simulations using a fully coupled atmosphere-ocean-wave model[J]. Ocean Modelling, 2018, 131: 132-151.

Other Comments:

L14: Rossby number (Ro = relative vorticity/Coriolis parameter);

**Response:** Thanks, we have added this defination.

L16: Rossby number;

**Response:** Thanks you, it have been corrected。

L166: vertical feedback of the ocean by ... Kalmaegi: Not sure what this means, what "feedback", maybe "response..."?

**Response:** Sorry for misunderstanding. It is proper to use "response", we have replaced it.

Also: I assume GLORY assimilates Argo data but not the Station data. If so, then it is unsurprising that GLORY agrees with ARGO but not Station 5 (Figure 1).

Only Station 5 on the left side of the storm was used to support the authors' hypothesis. To support (refute?) the Authors' hypothesis I suggest using data from other Stations (except #3), right and left of Kalmaegi.

**Response:** Due to the lack of temperature data at S1, we added the vertical profiles of S2 and S4 were compared with GLORYS12v1. The RMS between GLORYS12V1 and Station 2 (Station 4) is 0.14 (0.10), with significant deviations in the mixed layer and thermocline. Although compared to S5, the RMS of S2 and S4 is a little larger, but still acceptable.

[Figure]

**Figure S5.** Evaluation of GLORYS12V1 data performance during September 2014. (a) Vertical monthly mean temperature within the anticyclonic eddy AE2 (119.5°E 17.9°N) as measured by Station 2 (115.5°E 18.2°N) . (b) Comparison of vertical monthly mean temperature recorded at Station 4 (117.5°E 19.2°N).

L223: ... weak wind stress curl, to be more precise. The term "wind shear" is also customarily taken as "vertical wind shear" in TC studies in meteorology so can be confusing.

**Response:** Thank you, the word "vertical" has been added to be more precise.

L245: ... with 6-hourly dots.

**Response:** Thank you, the word "6-hourly" has been added to be more precise.

**All of the co-authors are so grateful to the reviewer for the time spent on our manuscript. The comments and suggestions provided by the reviewer are invaluable for us to improve our manuscript. We are so appreciated.**

---

## Author Comment (AC2)

**Response to Reviewer 2**

General comments:

The authors investigated the responses of two warm eddies to a typhoon using observational and reanalysis data. There have been lots of efforts on eddy feedback to TC evolution, while our understanding of eddy response to TCs remains limited. The work is potentially interesting and contribute to broaden the knowledge of TC-eddy interaction, but I have some comments before the paper can be published. I recommend moderate to minor revision of the manuscript.

**Response: We would like to thank you for your careful reading, helpful comments, and constructive suggestions, which have significantly improved the presentation of our manuscript. We have carefully considered all comments from the reviewer and revised our manuscript accordingly. The manuscript has also been double-checked, the typos and grammar errors we found have been corrected. In the following section, we summarize our responses to each comment from the reviewer. We believe that our responses have well addressed all concerns from the reviewer. The changes are highlighted in the manuscript. Please see below, in blue and black, for a point-by-point response to the reviewer's comments and concerns. All page numbers refer to the revised manuscript file with tracked changes.**

Primary comments:

1. The motivation of conducting the study should be more clarified. Just stating that "However, there has been relatively limited exploration of different responses exhibited by warm eddies under the influence of typhoons" is not adequate from my opinion. There have been several studies on the eddy response to TCs. The authors should be more carefully summarize what have been reported from these previous studies, and what new knowledge will be reported in this study.

**Response :** Thank you for your nice comments on our manuscript. According to your suggestions, we re-summarized what has been reported in previous studies of eddy-typhoon interactions, as well as what new knowledge will be reported in this study. please check Lines 98.

"Previous studies on the interaction between eddies and typhoons have primarily focused on two aspects, one is the influence of ocean eddies on typhoons: the enhancement of warm eddies on typhoons; and the other is how typhoons affect ocean eddies: the response of cold eddies to typhoons. However, the exploration of the response of warm eddies under the influence of typhoons and the three-dimensional response of eddies to typhoons is relatively limited. In this paper, we explore the effects of the different positions of warm eddies on the response of eddies as well as the changes in their three-dimensional thermohaline structure characteristics, which will provide inputs to the study of eddy-typhoon interactions."

2. The writing of the paper needs a substantial improvement. There are lots of sentences that are hard to follow and hinder understanding. The logics between paragraphs should be clear. On

Line 31, the word of "typhoon" is used, but in the following the "tropical cyclone" is used instead. The paper should keep consistency in word usage.

**Response :** Thanks for your suggestion. As some papers and also our study show that some tropical storm also play important role in air-sea interaction. So we change the word "typhoon" to "Tropical cyclones (TCs)" in Line 31. We also keep consistency in the revised manuscript.

Specific comments:

1. Is it reasonable to use the rectangle areas represent AE1 and AE2 (Figure 1)? As eddies always move during typhoon.

**Response :** Thank you for your asking. We have figured the spatial distribution of SLA and geostrophic current from 11 September to 25 September, and both AE1 and AE2 were active within the rectangular frames during this period. Here, we just show two days' snapshots, on 25 September (right panel), these two eddies still in the rectangular frames. Moreover, the moving speed of the two eddies from 11 September to 25 September was calculated through the "Mesoscale Eddy Trajectory Atlas" product. The mean moving speed of AE1 is 0.18 m s$^{-1}$, and 0.07 m s$^{-1}$ for AE2 during this period, so the mean travelling distance is about 217.7 km and 84.7 km, respectively, which are still in the rectangular frames during this period. Further more, we focus on the fixed area are helpful for comparison. So it is reasonable to use rectangular areas to represent the eddies during the study.

[Figure]

2.There are several common issues with the figures in the manuscript that need to be addressed, including the addition of x-labels and y-labels, as well as the unification of font sizes etc. Specific comments are as follows:

1). Figure 1: The first letter of "depth" in Y-label needs to be capitalized.

**Response:** Thanks, the figure has been modified as suggested.

2). Figure 4: Just keep one arrow legend and text of "15 m/s" in (a), the quiver and the text should be larger.

**Response:** Thanks, the figure has been modified as suggested.

3). The coordinate axes are duplicated in Figure 5, delete the x-axis and y-axis in Figure 5 (a), (d), (g), (j), just like Figure 4; Set the range of SST colorbar as 26 to 31 °C; Change the red dots to larger black dots.

**Response:** Thanks, the figure has been modified as suggested.

4). Figure 6: The first letter of "date" needs to be capitalized, like "Date".

**Response:** Thanks, the figure has been modified as suggested.

5). Figure 7: The first letter of "date" and "depth" needs to be capitalized; The unit of "psu" should be "PSU".

**Response:** Thanks, the figure has been modified as suggested.

6). Figure 8: The first letter of "date" and "depth" needs to be capitalized; Please use the density excess replace the density, that means density minus 1000 kg. m$^{-3}$; The unit of buoyancy should be written as "N$^2$ (10$^{-4}$ s$^{-2}$)".

**Response:** Thanks, the figure has been modified as suggested.

7). Figure 9: The first letter of "depth" needs to be capitalized; "psu" should be "PSU".

**Response:** Thanks, the figure has been modified as suggested.

8). Figure 10: The unit of wind stress curl unit should be "N.m$^{-3}$", so the colorbar will make some confuse, move "N.m$^{-3}$" to the right side of colorbar or another proper place. To compare AE1 and AE2, the same figure as Figure 10 but for AE2 is needed.

**Response:** Thanks, the figure has been modified as suggested. The similar figure for AE2 has been added and the corresponding sentences have been added at Line xxx.

[Figure]

9). Figure 11: The first letter of "date" and "depth" needs to be capitalized.

**Response:** Thanks, the figure has been modified as suggested.

3. There are some spelling or inappropriate use of singular and plural in the manuscript, for example,

L39 on 'the' one hand

L65 the typhoon track 'are' more intensely

L87 of a 'near-inertial' wake

L118 The daily Sea Level Anomaly (SLA) and geostrophic current data 'are' provided by Archiving

L131 'database'

L156 temperature and salinity 'from' 1 September to 30 September 2014 'were' chosen to study.

L216 'where'

L278 'passage'

L398 warm anomaly of 1.2 ℃ 'was' observed at a depth

L444 'Compared'
L508 'contributes'

**Response:** Thank you for your carefully reading and corrention.In our resubmmitted manuscript, the typos are revised at Line 39, 65, 87, 118, 131, 156, 216, 278, 398, 444 and 508, accordingly.

4. Inconsistent use of tenses. When describing the work of previous researchers, you should generally use the **past tense**. This is because those studies have been completed in the past and form part of the background for your research. When describing your own work, you should generally use the **present tense**. This helps emphasize that your results are current and still valid.
For example, L122 the data access needs to use the past tense. ?
L309-313 please use the past tense
L316-319 please use the present tense, etc?
Please check the full manuscript carefully.

**Response :** Thank you for your suggestions. We have corrented tense misuses in our new manuscript.

5. Many abbreviations have repeated definitions, including but not limited to:
L71 and L118 SLA duplication definitions.
'EPV' is firmly defined 4-5 times, 'Rossby number', 'SST', 'EKE', etc.
**Response:** Thanks. All abbreviations are only defined at the first time.

6. A mix of American and British spellings, such as Typhoon 'center' and 'centre' are appeared in the manuscript.
**Response:** Thanks for your careful checks. We have corrected the 'centre' to 'center' to make the word harmonized within the whole manuscript.

7. Line 650: Please check out the format of the reference.
**Response:** Thank you for your suggestion. We have checked the format of the reference at Line 650, it is correct. Please see the screenshot of this reference we cited.

**Effects of a Warm Oceanic Feature on Hurricane Opal**

Lynn K. Shay, Gustavo J. Goni, and Peter G. Black

Print Publication: 01 May 2000

DOI: https://doi.org/10.1175/1520-0493(2000)128<1366:EOAWOF>2.0.CO;2

Page(s): 1366–1383

**All of the co-authors are so grateful to you for the time spent on our manuscript. The comments and suggestions provided by the reviewer are invaluable for us to improve our**

**manuscript. We are so appreciated.**

---

## Author Response (AR4)

**Response to Reviewer 1**

General Comments:

The authors have followed the reviewers' comments to modify their manuscript, and hence the quality of the paper is much improved. the authors proofread their paper much more carefully before the paper can be accepted for publication.

Response: We express our gratitude for your thorough review, valuable comments, and constructive suggestions during your second review. Your input has greatly enhanced the clarity of our manuscript. We have meticulously reviewed all comments provided by the reviewer and have made revisions accordingly.

In the subsequent section, we summarize our responses to each comment from the reviewer. We believe that our responses have well addressed all concerns from the reviewer. The changes are highlighted in the manuscript. Please see below, in blue and black, for a point-by-point response to the reviewer's comments and concerns. All page numbers refer to the revised manuscript file with tracked changes.

Primary comments:
1. The previous primary comment 1 was not addressed properly. Although they have re-organized their rationality for conducting the work in the response letter, there seems no change in their modified manuscript. The authors seem not careful in preparing their response letter (There is even words like "Line xxx")

Response: We sincerely apologize for the notable error like "Line xxx" in our previous version. In this revised manuscript, we are taking our revised manuscript more carefully and conducted a final thorough check to rectify any identified typos and grammar errors. Considering the primary comment 1, we have reworked specific sections of introduction (Lines 91-96) and summary (Lines 537-547) to emphasize the innovations and research motivations. About influence of near-inertial energy, we have incorporated additional sentences at Lines 325-328. Furthermore, the discussion on Ekman layer depth has been expounded upon in the Discussion section (Lines 423-434).

2. There are numerous grammar or tense mistakes, especially in their newly supplemented sentences (e.g., Lines 25-27, 96-98, 321-322, Lines 414-415, Lines 418-431, etc).

Response: Thank you very much! We have carefully reviewed and made necessary corrections, which are highlighted in yellow in the revised manuscript. We have converted the entire manuscript to present tenses, including Lines 25-27, 96-98, 321-322, 414-415, 418-431 as your suggestion in the original manuscript.

Lines 25-26: we have rectified "triggered" into "triggers", "enhanced" into "enhances" .

Lines 409: we have adjusted "will delve" to "delve".

Lines 412-428: "The EPV is very small before the typhoon, measuring less than $0.5\times10^{-5}$ m s$^{-1}$ in both AE1 and AE2. However, during 15-16 September (Fig. 9c-f), when the typhoon crosses the NSCS, the EPV undergoes significant changes. Its absolute value increases to over $1.5\times10^{-4}$ m s$^{-1}$ within both AE1 and AE2. AE1 consistently exhibits a predominantly negative EPV during most of this period. Consequently, during Typhoon Kalmaegi, the negative EPV facilitates downwelling and convergence (Jaimes and Shay, 2015), leading to a warmer and fresher subsurface layer in AE1 (Fig. 6 a-b).

On the other hand, AE2 displays a more fluctuating pattern. It is positive on 14 September, shows both positive and negative values at 0000 UTC on 15 September, and remains mainly negative from 15 to 16 September, and eventually returning to positive, reflecting a continuously fluctuating process. The positive EPV in AE2 contributes to the influx of colder subsurface water into the upper layers, resulting in surface and subsurface water cooling and an increase in salinity in the subsurface (Fig. 6c-d). Correspondingly, the variations in Ekman layer depth ($D_E$) with the typhoon's passage are similar to EPV, as shown in Fig. 10. When Kalmaegi approaches at 0000 UTC on 14 September, the mean $D_E$ within AE1 is only 21 m, while in AE2, it is 114 m. This indicates that AE2 has already been influenced by Typhoon Kalmaegi. Subsequently, the depth of the $D_E$ within AE2 sharply deepens, reaching its maximum depth of 241 m at 0000 UTC on 15 September, coinciding with the proximity of Typhoon Kalmaegi's center to AE2."

All of the co-authors are so grateful to you for the time spent on our manuscript. The comments and suggestions provided by the reviewer are invaluable for us to improve our manuscript. We are so appreciated.

**Response to Reviewer 2**

General comment:

The authors examined the response of two pre-existing warm eddies to Typhoon Kalmaegi based on observations and reanalysis data. After the typhoon's passage, the two warm eddies presented different changes in terms of amplitude, Rossby number, and kinetic energy. The authors ascribed this difference to the relative positions of warm eddies to the typhoon. I recommend acceptance of the manuscript after a minor revision.

Response: We would like to thank you for your careful reading, helpful comments, and constructive suggestions, which have significantly improved the presentation of our manuscript. We have carefully considered all comments from the reviewer and revised our manuscript accordingly. In the following section, we summarize our responses to each comment from the reviewer. We believe that our responses have well addressed all concerns from the reviewer. The changes are highlighted with yellow in the manuscript. Please see below, in blue and black, for a point-by-point response to the reviewer's comments and concerns. All page numbers refer to the revised manuscript file with tracked changes.

Minor comment:
1. Lines 88-89: The focus of this study is the relative locations of two warm eddies to the typhoon center. Therefore, the maximum wind radius of Typhoon Kalmaegi is an essential metric and must be stated clearly, especially the maximum wind radius when Kalmaegi passed AE1 and AE2.

Response: Your comments have been immensely benefici al, and we sincerely appreciated! Consequently, we have added the marking of the one- and two-time maximum wind radius of Typhoon Kalmaegi in Figure 3. Furthermore, we have provided explanations regarding the maximum wind radius and relative distance between Typhoon and eddies when it passed AE1 and AE2 in the revised manuscript. Kindly review the revisions made in Lines 228-231, Lines 232-233, Lines 443-451, 467-471.

[Figure]

**Figure 3.** The variations in sea level anomaly before and after Typhoon Kalmaegi moved over the anticyclonic eddies AE1 and AE2 between 14 September and 19 September **(a-f)**. The black solid rectangle represents the area of AE1, while the black dashed rectangle represents the area of AE2. The blue solid line depicts the path of typhoon Kalmaegi, and the solid red and dashed blue circles are one- and two-times the maximum wind radius of the typhoon, while the blue dotted line in **(f)** is the path of tropical storm Fung-wong (best-track data sourced from CMA).

Lines 228-231: "Throughout this intensification stage, Kalmaegi encounters two warm eddies: anticyclonic eddy AE1, is positioned to the left of the typhoon's path, with its core situated on the periphery of the typhoon's two-times maximum wind radius (Fig.3c-d)."

Lines 232-233: "AE2 precisely intersects with the typhoon's trajectory, and its core nearly coincides with the maximum wind radius of the typhoon (Fig.3b-d)."

Lines 443-451: "After traversing the warm ocean characteristics of AE2, Typhoon Kalmaegi strengthens, resulting in a reduction of the maximum wind radius. As it passed through AE1, the maximum wind radius is 35 km. Notably, the center of AE1 is located outside the typhoon's two-times maximum wind radius, approximately 104 km away from the typhoon center (Fig. 3). As mentioned earlier, strong upwelling occurs within two-times maximum wind radius, while weak subsidence exists in the vast area outside the upwelling region (Jaimes and Shay, 2015). Hence, the hypothesis presented here suggests that the observed intensification of AE1 on the left side of the typhoon track is more likely attributed to the negative wind stress generated outside the maximum wind radius, driving the enhancement of downwelling in the pre-existing anticyclonic feature in the ocean."

Lines 467-471: "The response of AE2 differs from that of AE1 mainly because AE2 is quite near the Typhoon Kalmaegi's track. As the typhoon passes through AE2, the maximum wind radius is 48 km. AE2 is merely 26 km away from the typhoon center (Fig. 3). The significantly positive wind stress curl at the typhoon center induces upwelling and positive vorticity downward into the eddy (Huang and Wang, 2022), and noticeably weakens the eddy, corresponding to the decrease in SLA (Fig. 12a)."

All of the co-authors are so grateful to the reviewer for the time spent on our manuscript. The comments and suggestions provided by the reviewer are invaluable for us to improve our manuscript. We are so appreciated.

**Response to Reviewer 3**

General comment:
I am not sure about the novelty of the results presented here. The aim of the paper is not presented clearly. As a consequence, it is difficult for the reader can see the value of the manuscript in the context of an already very rich literature on the subject. The manuscript requires a major revision.

Response: We express our sincere gratitude for your thorough review, valuable comments, and constructive suggestions, all of which have significantly enhanced the quality of our manuscript. We have diligently addressed your comment and made corresponding revisions to improve clarity and accuracy. The manuscript has undergone a meticulous double-check, ensuring that identified typos and grammar errors have been rectified. These changes are highlighted by yellow in the revised manuscript. All page numbers refer to the revised manuscript with tracked changes.

We particularly appreciate your regarding the clarity of our introduction. In response, we have strengthened both the introduction (Lines 91-96) and summary (Lines 537-547) sections to better elucidate the novelty and purpose of our research.

Primary comments:
1、I am not sure about the novelty of the results presented here.
Response: Thanks for your suggestion. While many previous studies have explored the interaction between TCs and eddies and have drawn generalized conclusions, such as the weakening (strengthening) effects of cold (warm) eddies on TCs and TCs are recognized for strengthening cold eddies and weakening warm eddies, our study takes a different approach. We aim to illustrate that even when TCs encounter eddies of the same polarity, the response of these eddies to TCs exhibits variations. This nuanced response is intricately linked to factors including the relative position of the eddies and the TCs, the eddies' intensity, and the background current. Notably, this is the first time it has been discussed in the South China Sea. By analyzing wind stress curl distribution, EPV, buoyancy frequency and the relative position between the eddies and the typhoon's track, this case study provides a more nuanced understanding of the mechanisms driving these different eddy-TC interactions in the Northern South China Sea. Moreover, it will further improve the accuracy of TC forecasts and enhancing the simulation capabilities of air-sea coupled models.

We have re-written this part in Lines 537-547.

2、The aim of the paper is not presented clearly.
Response: Thank you so much. The NSCS frequently experiences intense tropical cyclones (TCs), coinciding with notable mesoscale eddies activity in the region.

Following the passage of a TC, the alteration or intensification of mesoscale eddies exerts an impact on the subsequent TC wake response and geostrophic adjustment process. This, in turn, leads to variations in ocean temperature and salt distribution in the local regions of the TC channel. Additionally, it influences the air-sea interaction of tropical cyclones following similar paths, a factor crucial for the accurate prediction of the next TC. Based on in-situ datasets, multi-platform satellite measurements, and GLORYS12V1 reanalysis data, we investigate the influence of two anticyclonic eddies on upper ocean responses to Typhoon Kalmaegi. This marks our initial endeavor to characterize the distinct physical variations induced by TCs within two eddies of the same polarity. This effort contributes to a deeper understanding of the role played by mesoscale eddies in modulating interactions between TCs and the ocean, and a more detailed understanding of the driving mechanisms of eddy-TC interactions in the northern South China Sea.

We have re-written this part in Lines 91-96.

All of the co-authors are so grateful to the reviewer for the time spent on our manuscript. The comments and suggestions provided by the reviewer are invaluable for us to improve our manuscript. We are so appreciated.

---

## Author Response (AR5)

**Response to Reviewer 1**

General Comments:

The authors have followed the reviewers' comments to modify their manuscript, and hence the quality of the paper is much improved. the authors proofread their paper much more carefully before the paper can be accepted for publication.

**Response:** We express our gratitude for your thorough review, valuable comments, and constructive suggestions during your second review. Your input has greatly enhanced the clarity of our manuscript. We have meticulously reviewed all comments provided by the reviewer and have made revisions accordingly.

In the subsequent section, we summarize our responses to each comment from the reviewer. We believe that our responses have well addressed all concerns from the reviewer. The changes are shown in the manuscript. Please see below, in blue and black, for a point-by-point response to the reviewer's comments and concerns. All page numbers refer to the revised manuscript file with tracked changes.

Primary comments:

1. The previous primary comment 1 was not addressed properly. Although they have re-organized their rationality for conducting the work in the response letter, there seems no change in their modified manuscript. The authors seem not careful in preparing their response letter (There is even words like "Line xxx")

**Response:** We sincerely apologize for the notable error like "Line xxx" in our previous version. In this revised manuscript, we are taking our revised manuscript more carefully and conducted a final thorough check to rectify any identified typos and grammar errors. Considering the primary comment 1, we have reworked specific sections of introduction (Lines 88-93) and summary (Lines 536-546) to emphasize the innovations and research motivations. About influence of near-inertial energy, we have incorporated additional sentences at Lines 411-438. Furthermore, the discussion on the influence of background flow field has been expounded upon in the Discussion section (Lines 439-449).

2. There are numerous grammar or tense mistakes, especially in their newly supplemented sentences (e.g., Lines 25-27, 96-98, 321-322, Lines 414-415, Lines 418-431, etc).

**Response:** Thank you very much! We have carefully reviewed and made necessary corrections, which are shown in the revised manuscript. We have converted the entire manuscript to present tenses, including Lines 25-27, 96-98, 321-322, 414-415, 418-431 as your suggestion in the original manuscript.

Lines 25-26: we have rectified "enhanced" into "enhances".

Lines 408: we have adjusted "will delve" to "delve".

Lines 428-438: "The EPV is very small before the typhoon, measuring less than $0.5 \times 10^{-5}$ m s$^{-1}$ in both AE1 and AE2. However, during 15-16 September (Fig. 9c-f), when the typhoon crosses the NSCS, the EPV undergoes significant changes. Its absolute value increases to over $1.5 \times 10^{-4}$ m s$^{-1}$ within both AE1 and AE2. AE1 consistently exhibits a predominantly negative EPV during most of this period. Consequently, during Typhoon Kalmaegi, the negative EPV facilitates downwelling and convergence (Jaimes and Shay, 2015), leading to a warmer and fresher subsurface layer in AE1 (Fig. 6 a-b). On the other hand, AE2 displays a more fluctuating pattern. It is positive on 14 September, shows both positive and negative values at 0000 UTC on 15 September, and remains mainly negative from 15 to 16 September, and eventually returning to positive, reflecting a continuously fluctuating process. The positive EPV in AE2 contributes to the influx of colder subsurface water into the upper layers, resulting in surface and subsurface water cooling and an increase in salinity in the subsurface (Fig. 6c-d).

All of the co-authors are so grateful to you for the time spent on our manuscript. The comments and suggestions provided by the reviewer are invaluable for us to improve our manuscript. We are so appreciated.

**Response to Reviewer 2**

General comment:

The authors examined the response of two pre-existing warm eddies to Typhoon Kalmaegi based on observations and reanalysis data. After the typhoon's passage, the two warm eddies presented different changes in terms of amplitude, Rossby number, and kinetic energy. The authors ascribed this difference to the relative positions of warm eddies to the typhoon. I recommend acceptance of the manuscript after a minor revision.

**Response:** We would like to thank you for your careful reading, helpful comments, and constructive suggestions, which have significantly improved the presentation of our manuscript. We have carefully considered all comments from the reviewer and revised our manuscript accordingly. In the following section, we summarize our responses to each comment from the reviewer. We believe that our responses have well addressed all concerns from the reviewer. The changes are shown in the manuscript. Please see below, in blue and black, for a point-by-point response to the reviewer's comments and concerns. All page numbers refer to the revised manuscript file with tracked changes.

Minor comment:

1. Lines 88-89: The focus of this study is the relative locations of two warm eddies to the typhoon center. Therefore, the maximum wind radius of Typhoon Kalmaegi is an essential metric and must be stated clearly, especially the maximum wind radius when Kalmaegi passed AE1 and AE2.

**Response:** Your comments have been immensely beneficial, and we sincerely appreciated! Consequently, we have added the marking of the $R_{max}$ of the typhoon and width of typhoon-induced baroclinic geostrophic response in Figure 3. Furthermore, we have provided explanations regarding the maximum wind radius ($R_{max}$) and when it passed AE1 and AE2 in the revised manuscript. Kindly review the revisions made in Lines 226-229, Lines 230-231, Lines 419-427, Lines 472-476.

[Figure]

**Figure 3.** The variations in sea level anomaly before and after Typhoon Kalmaegi moved over the anticyclonic eddies AE1 and AE2 between 14 September and 19 September **(a-f)**. The black solid rectangle represents the area

of AE1, while the black dashed rectangle represents the area of AE2. The blue solid line depicts the path of typhoon Kalmaegi, and the solid red and dashed blue circles are the one-times $R_{max}$ of the typhoon and width of typhoon-induced baroclinic geostrophic response, while the blue dotted line in **(f)** is the path of tropical storm Fung-wong (best-track data sourced from CMA).

Lines 226-229: "Throughout this intensification stage, Kalmaegi encounters two warm eddies: anticyclonic eddy AE1, is positioned to the left of the typhoon's path, with its core situated on the periphery of the typhoon's one-times $R_{max}$ (Fig.3c-d)."

Lines 230-231: "AE2 precisely intersects with the typhoon's trajectory, and its core nearly coincides with the $R_{max}$ of the typhoon (Fig.3b-d)."

Lines 419-427: "Most of the positive wind stress curl exists within $R_{max}$, leading to strong upwelling, while the weak negative wind stress curl occurs outside R_max, resulting in weak subsidence caused by TCs exist outside the upwelling area (Lu et al., 2020; Lu and Shang, 2024). Typhoon Kalmaegi strengthened after passing through the warm ocean characteristics of AE2, causing a reduction in $R_{max}$. When passing AE1, $R_{max}$ is 37 km. Notably, the center of AE1 is located outside the $R_{max}$ (Figure 3). Hence, the hypothesis presented here suggests that the observed intensification of AE1 on the left side of the typhoon track is more likely attributed to the negative wind stress curl generated outside the $R_{max}$, thereby driving the enhancement of downwelling in the pre-existing anticyclonic feature in the ocean."

Lines 472-476: "The response of AE2 differs from that of AE1 mainly because AE2 is quite near the Typhoon Kalmaegi's track. As the typhoon passes through AE2, the $R_{max}$ is 46 km. AE2 is merely 26 km away from the typhoon center (Fig. 3). The significantly positive wind stress curl at the typhoon center induces upwelling and positive vorticity downward into the eddy (Huang and Wang, 2022), and noticeably weakens the eddy, corresponding to the decrease in SLA (Fig. 12a)."

All of the co-authors are so grateful to the reviewer for the time spent on our manuscript. The comments and suggestions provided by the reviewer are invaluable for us to improve our manuscript. We are so appreciated.

**Response to Reviewer 3**

General comment:

I am not sure about the novelty of the results presented here. The aim of the paper is not presented clearly. As a consequence, it is difficult for the reader can see the value of the manuscript in the context of an already very rich literature on the subject. The manuscript requires a major revision.

**Response:** We express our heartfelt thanks for your thorough review, valuable comments, and constructive suggestions, all of which have significantly enhanced the quality of our manuscript. We have diligently addressed your comment and made corresponding revisions to improve clarity and accuracy. The manuscript has undergone a meticulous double-check, ensuring that identified typos and grammar errors have been rectified. These changes are shown in the revised manuscript. All page numbers refer to the revised manuscript with tracked changes.

An important issue is that the response made to reviewer 1 does not seem adequate. Reviewer 1 criticizes your interpretation: he says that because the typhoon is moving too fast for the wind stress curl to have a direct influence, all the influence of the typhoon should occur through vertical mixing driven by near-inertial waves, which is different in anticyclones compared with cyclones (see for example Jaimes et al 2011, Journal of Physical Oceanography). If I read correctly your answer to the reviewer, you claim that the wind stress changes, so the Ekman depth changes: but the Ekman depth is a measure of how deep the mixing is, not a measure of the geostrophic response to the wind stress curl. The standard (textbook) definition of the Ekman depth depends on the vertical mixing coefficient and the Coriolis frequency, it does not depend on the wind stress nor on the wind stress curl. Therefore you do not answer the reviewer question. An interesting reference to answer reviewer 1 could be Lu et al (JGR, 2023), who demonstrate the influence of the wind stress curl on the eddies. Or, perhaps, you could make an answer based on Jaimes and Shay (2015), because those authors study the response of warm eddies to a hurricane, just like you do; note that they point out that it is not the hurricane wind stress curl ("undisturbed" Ekman pumping) which is important but rather the "nonlinear" Ekman pumping (second term in their equation 5). You would need to estimate parameters similar to the ones of Lu et al, or Jaimes and Shay, for the specific case of your warm eddies and your typhoon Kalmaegi, to be able to argue that either of these interpretations applies to your case.

**Response:** Thank you for your suggestions and recommended papers.

The response caused by TCs in the ocean is not only near-inertial response, but also geostrophic response. The near-inertial response influences the large-scale and mesoscale ocean circulation through vertical mixing driven by near-inertial waves. However, geostrophic response is induced by all TCs but near-inertial response only can be done for $U_h > C$, where $U_h$ is the moving speed of a TC and $C$ is the baroclinic mode wave speed (Lu et al., 2023). The potential vorticity injected by typhoon leads to

quasi-geostrophic adjustment of eddy, and the potential vorticity anomalies caused by wind stress curl are generated by geostrophic response (Lu et al., 2020). In addition, the geostrophic response of typhoon is generated within about 0.5 day, and we believe that the wind stress curl of Typhoon Kalmaegi has an impact. The Ekman depth does depend on the vertical mixing coefficient and the Coriolis frequency. In this paper, the Ekman depth is calculated by $D_E = \frac{7.12}{\sqrt{\sin|\varphi|}} U_{10}$(Li et al.,2022), where $U_{10}$ represents the wind speed at 10 m above the sea, and $\varphi$ is the latitude. Therefore, we removed the discussion on the Ekman depth.

Li, Y., Yang, D., Xu, L., Gao, G., He, Z., Cui, X., et al. (2022). Three types of typhoon-induced upwellings enhance coastal algal blooms: A case study. Journal of Geophysical Research: Oceans, 127, e2022JC018448. https://doi.org/10.1029/2022JC018448

Lu, Z., G. Wang, and X. Shang, 2023: Observable Large-Scale Impacts of Tropical Cyclones on the Subtropical Gyre. *J. Phys. Oceanogr.*, **53**, 2189–2209, https://doi.org/10.1175/JPO-D-22-0230.1.

Regarding the parameters in the relevant papers you mentioned, we get the following results:

TCs influence mesoscale eddies through baroclinic geostrophic response (Lu et al., 2020). The width of this response is generally constrained within the TC orbit, with the transverse diameter length represented as $L_h = L_d + R_{max}$ (Lu and Shang, 2024). Here, $L_d$ is the first mode of Rossby deformation radius, and $R_{max}$ denotes the maximum wind radius. $L_d = \frac{c}{f}$, the phase speed of the first baroclinic mode $c$ was obtained using the method outlined in Jaimes and Shay (2009). Therefore, the width of Typhoon Kalmaegi-induced baroclinic geostrophic response falls within the range of 92 km (Figure 3). Essentially, these geostrophic effects are caused by wind stress curl, and wind stress curl injects disturbance into the ocean through upwelling and downwelling. Most of the positive wind stress curl exists within $R_{max}$, leading to strong upwelling, while the weak negative wind stress curl occurs outside $R_{max}$, resulting in weak subsidence caused by TCs exist outside the upwelling area (Lu et al., 2020; Lu and Shang, 2024). Typhoon Kalmaegi strengthened after passing through AE2, causing a reduction in $R_{max}$. When passing AE1, $R_{max}$ is 37 km. Notably, the center of AE1 is located outside the $R_{max}$ (Figure 3). Hence, the hypothesis presented here suggests that the observed intensification of AE1 on the left side of the typhoon track is more likely attributed to the negative wind stress curl generated outside the $R_{max}$, thereby driving the enhancement of downwelling in the pre-existing anticyclonic feature in the ocean. (Lines 411-427)

Considering the influence of the background flow field, the pumping rate $W$ is not only related to the wind stress curl (undisturbed Ekman pumping), but also related to the curl of background geostrophic flow (nonlinear Ekman pumping). Therefore, in order to describe the response of upwelling and downwelling more accurately, a parametric TC-driven pumping velocity scale $W_s = W_E - R_o\delta(U_h + U_{OML})$ (Jaimes

and Shay, 2015), is derived from the time-dependent vorticity balance in the ocean mixed layer. Here, $W_E$ calculated by Eq. (8), $R_o$ is calculated using Eq. (3), the aspect ratio is calculated by $\delta = \frac{h}{R_{max}}$, here $h$ represents oceanic mixed layer thickness, $U_h$ denotes the translation speed. The oceanic mixed layer Ekman drift is calculated by $U_{OML} = \frac{\tau R_{max}}{\rho h U_h}$. The vertical velocity $W_s$ calculated by Eq. (11) are presented in Figure 10. When Typhoon Kalmaegi passes through AE1, the $W_s$ in AE1 obviously increases, while AE2 experiences minimal change. (Lines 439-449)

[Figure]

**Figure 10**. TC-driven pumping velocity ($W_s$) from 14 September to 16 September (a-f). The color represents the $W_s$, the blue solid line is the path of Kalmaegi. Negative and positive values are for upwelling and downwelling regimes, respectively.

The introduction could be improved. The literature review is too long and not very easy to read. The lines 96 to 98 are not enough to motivate the paper: reading this sentence, it seems that you just want to add two more eddies to the already rich literature on eddy-cyclone interactions, without asking specific scientific questions, nor pointing out why your study is really original and important.

**Response:** We are especially grateful for your regarding the clarity of our introduction. In response, we have rewritten the introduction, and strengthened both the introduction (Lines 88-93) and summary (Lines 536-546) sections to provide a clearer elucidate the novelty and purpose of our research.

In the introduction you don't justify at all why you consider typhoon Kalmaegi. How special is this typhoon, relative to other TCs in the area? In the response to the reviewers you mention that Zhang has published 6 other papers about this typhoon and the observations made in 2014, but this is not highlighted in the introduction. You need to explain better why the present paper is different from the work that has already been published, what is new here. Are the observations you show already published elsewhere? If it is the case, why is another paper warranted?

**Response:** Typhoon Kalmaegi passed over an array of buoys and moorings in the northern South China Sea during September 2014, leaving a set of observations on typhoon-induced dynamical and thermohaline responses of the upper ocean. Therefore,

Zhang (2016, 2018) conducted research on the upper ocean's responses to typhoon Kalmaegi. Concurrently, we observed that the typhoon also encountered two warm eddies, each exhibiting distinct responses to the typhoon. Using multi-source data, we investigate how two anticyclonic eddies respond to Typhoon Kalmaegi. Thus, this study forces on understanding the response of eddies to the typhoon, rather than solely examining the upper ocean. This marks the initial effort to characterize the different physical variations induced by TCs within two same polarity eddies, contributing to a better understanding of the role played by mesoscale eddies in modulating interactions between TCs and the ocean.

Zhang H, Wu R, Chen D, Liu X, He H, Tang Y, Ke D, Shen Z, Li J, Xie J, Tian D, Ming J, Liu F, Zhang D, Zhang W. Net Modulation of Upper Ocean Thermal Structure by Typhoon Kalmaegi (2014). Journal of Geophysical Research: Oceans, 2018, 123(10): 7154-7171.

Zhang H, Chen D, Zhou L, Liu X, Ding T, Zhou B. Upper ocean response to typhoon Kalmaegi (2014). Journal of Geophysical Research: Oceans, 2016, 121(8): 6520-6535.

line 417 and following: The discussion is very descriptive and does not discuss what is new in your results compared with the literature. The typhoon has a different effect on the two eddies. Is this just what one would expect based on the different positions of the eddies relative to the typhoon track, based on the existing literature? Or is there a bit of surprise in your observations? The way the manuscript is written, without first laying out hypothesis and more precise scientific questions, it is difficult for the reader to understand whether you just confirm existing theories with additional observations (which is worthy in itself), or whether there is something new (which is more exciting). When you say "The negative wind stress curl induced by the typhoon resulted in favourable surface ocean currents that further enhanced the clockwise rotation of the warm eddy": are you sure this sentence is valid in view of the high translation speed of the typhoon (reviewer 1's remark?)

**Response:** Thanks to your suggestion, we have rewritten the discussion and proposed a hypothesis based on previous theories. Using multiple observations in the South China Sea, we demonstrate that eddies of the same polarity exhibit different responses to same typhoon. Factors such as the distance between eddies and typhoons, eddies intensity and the background field need to be considered. We are sorry for our misrepresentation, and have removed the sentence "The negative wind stress curl induced by the typhoon resulted in favourable surface ocean currents that further enhanced the clockwise rotation of the warm eddy". Please check Lines 411 to 449.

The summary just repeats the main elements of the discussion (different response of the two eddies) but it lacks perspectives.

**Response:** We have rewritten the conclusion and stated the purpose and perspectives of this paper. Please check Lines 512 to 546.

A few detailed remarks:
1.Lines 54 to 57: I don't understand this sentence. It does not seem to be grammatically

correct.

**Response:** We apologize for the language problems in Lines 54 to 57. In light of the introduction's revision, we have deleted this sentence. Our intention is to convey that " TCs cause the strengthening of cyclonic eddies, leading to positive potential vorticity anomalies".

2.line 59: "In general, TCs strengthen cold eddies": this statement seems in contradiction with Sun et al 2014, who say "only about 10% of COEs were significantly influenced by these super typhoons". It would be more appropriate to say "In some cases" " rather than "in general".

**Response:** Thanks for your suggestion. We have amended "In general" to "In some cases" in Lines 55 as your recommendation.

3.line 73: "reduction of warm eddies": to you mean a reduction in numbers (less eddies)? Or do you mean a weakening of each eddy?

**Response:** Thanks your for bringing the ambiguity in our original sentence. We have now revised the sentence as follows: "Generally, TCs lead to a weakening of warm eddies" in Lines 68.

4.line 96: "previous studies focused on the interaction of cold cyclonic eddies and TCs: is it true than warm eddies have been overlooked? You refer to many publications about the interaction with warm eddies (lines 73 to 95), how do you assess that the warm eddies have not been focused on? The second part of that sentence is not clear. What are you investigating? The effect of the typhoon on an eddy is not a "feedback", is it?

**Response:** We sincerely apologize for the inaccuracies in our description. Previous studies have predominantly focused on exploring the interaction between TCs and eddies, often leading to generalized conclusions, such as the weakening (strengthening) effects of cold (warm) eddies on TCs. However, limited researches have been conducted on the divergent responses of same polarity eddies induced by the same typhoon process, particularly in the South China Sea. Based on in-situ datasets, multi-platform satellite measurements, and GLORYS12V1 reanalysis data, we investigate how the upper ocean within two anticyclonic eddies responds to Typhoon Kalmaegi. This marks an initial effort to characterize the different physical variations induced by TCs within two same polarity eddies, contributing to a better understanding of the role played by mesoscale eddies in modulating interactions between TCs and the ocean. Therefore, we have rewritten this section accordingly. Please check Lines 88 to 93.

5.lines 171-173: this is an example of badly constructed sentences. There are many problems with grammar in the manuscript.

**Response:** We apologize for the language issues present in the original manuscript. This sentence has been revised as following "Since the GLORY.S12V1 data assimilates data from Argo floats, it demonstrates good agreement with Argo profiling floats". Meanwhile, we have thoroughly reviewed the entire manuscript and enlisted the assistance of a native speaker to aid in revising the manuscript.

6.Figure 2a is not informative at all. If GLORYS assimilated the ARGO data at that time and location, the comparison is not a validation of the product. I suppose that GLORYS did not assimilate the data from the buoys? Then Figure 2b shows a real validation with independant data. It would be better to show profiles at different buoy locations only, and not an ARGO profile, in Figure 2.

**Response:** The temperature and salinity data of GLORYS12 used for assimilation analysis come from Copernicus Marine CORAv4.1 database. The CORA observations come from many different sources collected by Coriolis data center in collaboration with the In Situ Thematic Centre of the Copernicus Marine Service (CMEMS INSTAC). The observation integrated data from different types of instruments, primarily including Argo floats, XBT, CTD and XCTD, and Moorings. As temperature data was unavailable at S1, we supplemented compared vertical profiles from S2 and S4 with GLORYS12v1. The root mean square (RMS) difference between GLORYS12V1 and Station 2 (Station 4) is 0.14 (0.10), with significant deviations in the mixed layer and thermocline. While the RMS for S2 and S4 is slightly higher compared to S5, it remains within an acceptable range. Please check Lines 156 to 166.

[Figure]

**Figure 2**. Evaluation of GLORYS12V1 data performance during September 2014. **(a)**, **(b)** and **(c)** are the comparison of vertical monthly mean temperatures recorded at stations 2(115.5°E 18.2°N), Station 4 (117.5°E 19.2°N) and Station 5 (117°E 17.7°N) respectively.

7.lines 195 to 199, EKE definition: you need to say relative to what (time mean? spatial mean?) the anomaly is computed. Also, you should say how you compute the Ekman depth.

**Response:** Following your suggestion, we have added this sentence "The geostrophic velocity anomalies are referenced to the period of 1993 to 2012."in Lines 191-192. We also added the formula and explanation for calculating the Ekman depth above as requested.

All of the co-authors are so grateful to the reviewer for the time spent on our manuscript. The comments and suggestions provided by the reviewer are invaluable for us to improve our manuscript. We are so appreciated.

---

## Author Response (AR6)

**Response to Corrections**

**Dear Anne Marie,**

**We would like to express our sincere gratitude for your outstanding work and professional dedication throughout the revision processes of our paper. Your diligence and contribution have been indispensable in each round of revisions. Your expert advice and review comments have provided valuable guidance and insights for our paper. Your meticulous scrutiny and constructive feedback have enabled us to continuously improve the paper, ensuring its academic rigor and quality. Your efforts have played a crucial role in supporting our research endeavors.**

**We are deeply appreciative of the effort and dedication you invest in your role as a journal editor, and we hold great respect for your professional competence and commitment. We have taken the comments carefully and have made corrections. The changes are shown in the manuscript. Please see below, in blue and black, for a point-by-point response to comments. All page numbers refer to the revised manuscript file with tracked changes.**

Abstract:
The use of tenses is not uniform. If you choose the present tense, say "AE2 (...) Rossby number and EKE decrease", not decreased.
AE2 weakens
**Response:** We have rectified "weaken" into "weakens", "decreased" into "decrease" in lines 16, 17.

line 260 " increases by 1.3 cm"
**Response:** We have rectified "increases 1.3 cm" into "increases by 1.3 cm" in lines 260, 17.

lines 260, 263: You say "area of the AE1" but "core of AE2": the use of "the" should be the same everywhere. You should remove "the" before the anticyclone names AE1 and AE2.
**Response:** We have deleted "the" before the anticyclone names AE1 and AE2 in the lines 260, 263.

line 326: "persistence"
**Response:** We have corrected "perisistence" to "persistence" in line 326.

line 445 "using"

**Response:** We have corrected "useing" to "using" in line 445.

line 447 "the vertical velocity... are presented": either the velocity is presented, or the velocities are presented.
**Response:** We have corrected "the vertical velocity... are presented" to "the vertical velocity... is presented" in line 447.

line 460: instead of "warm core of the eddy AE1" just say "warm core of AE1".
**Response:** We have deleted "the eddy" in line 460.

line 464: "the downwelling of warm eddy" is not grammatically correct. You could "enhances the downwelling and inputs negative vorticity in AE1".
**Response:** We have corrected "the downwelling of warm eddy" to "enhances the downwelling and inputs negative vorticity in AE1" in line 464.

line 503: rather say "within AE2".
**Response:** We have deleted "the" in line 502.

line 515: "locates" and "strengthens with amplitude" do not seem good style. You could say "AE1 is located outside (...). Its amplitude, Ro and EKE strengthen after the passage of the Typhoon. "
**Response:** We have rewritten this sentence as "AE1 is located outside (...). Its amplitude, Ro and EKE strengthen after the passage of the Typhoon." in lines 514, 515.

line 516 "weakens with amplitude, Ro and EKE" is not a good style. You could perhaps just say "weakens"?
**Response:** We have rewritten this sentence as "AE2 weakens, which positions within the $R_{max}$ of typhoon." in line 515.

line 527: I would suggest "AE1, located outside the $R_{max}$ of the typhoon, is subjected to a negative wind stress curl which generates a potential vorticity perturbation inside the eddy. " (It seems obvious you are talking about the typhoon, do you need to repeat typhoon three times?)
**Response:** We have rewritten this sentence as "AE1, located outside the $R_{max}$ of the typhoon, is subjected to a negative wind stress curl which generates a potential vorticity perturbation inside the eddy." in lines 526, 527.

line 529: "is enhanced"
**Response:** We have corrected "is enhances" to "is enhanced" in line 527.

line 530 "within AE1"
**Response:** We have deleted "the" in line 528.

line 535 "respond differently to the typhoon".

**Response:** We have corrected to "respond differently to the typhoon" in line 533.

line 536-537: If you use "while", you cannot stop the sentence at the end of line 537. A grammatically correct sentence could be "while..., conversely...). However, long sentences should be avoided: I think the best would be to remove "while" in line 536.
**Response:** We have deleted "While" in line 535.

line 538: "Conversely...." this sentence says exactly the same thing as the previous one: it should be removed to avoid repetition.
**Response:** We have deleted this sentence in line 536.

line 542: "This is discussed for the first time".
**Response:** We have corrected "It is discussed first time" to "This is discussed for the first time" in line 540.

line 546: "and enhance"
**Response:** We have corrected "and enhancing" to "and enhance" in line 544.

All of the co-authors are so grateful to you for the time spent on our manuscript. The comments and suggestions provided by the editor are invaluable for us to improve our manuscript. We are so appreciated.